

**Enhancements of Airborne Particulate Arsenic over the Subtropical**
**Free Troposphere in the Springtime: Impact by South Asian Biomass**
**Burning**
**Yu-Chi Lin[1,2], Shih-Chieh Hsu[2], Chuan-Yao Lin[2], Shuen-Hsin Lin[2], Yi-Tang**
**Huang[1], Yunhua Chang[1], Yan-Lin Zhang[1*]**
[1.] *Yale-NUIST Center on Atmospheric Environment, Nanjing University of Information*
*Science and Technology, Nanjing, Jiangsu, China.*
[2.] *Research Center for Environmental Changes (RCEC), Academia Sinica, Taipei,*
*Taiwan, R.O.C.*
*Corresponded to Yan-Lin Zhang (*zhangyanlin@nuist.edu.cn;
dryanlinzhang@outlook.com*)*
**ABSTRACT**
Arsenic (As) has long been recognized as a toxic element of mainly
anthropogenic origins, having adverse effects on human health. However, there is
insufficient understanding regarding As released into atmosphere from biomass
burning (BB). To this end, daily airborne As concentrations in total particulate matter
(TSP) were determined at Mount Hehuan (24.16°N, 121.29°E, 3001 m a.s.l.), Taiwan
from September 2011 to September 2012. During the sampling period, As
concentrations varied from 0.02 to 5.9 ng m$^{-3}$, with a mean value of $0.5 \pm 1.0$ ng m$^{-3}$.
Significant seasonality of As was found over the subtropical free troposphere, with a
maximum concentration in the springtime. Based on backward trajectory analyses and
WRF-Chem model simulations, we found the high As concentrations during the
spring period were attributed to the biomass burning activities over South (S) Asia
where ground water, soil and crops are severely contaminated by arsenic. A good



correlation ($r = 0.73$ $p < .05$) between As and potassium ion (K$^+$, a chemical tracer of
BB activities) in S Asian BB events also supported this hypothesis. During the S
Asian BB events, the high As/Pb ratios ($> 0.2$) were also observed, indicating that
burning crops contaminated by lead arsenate could be a crucial candidate for
extremely high As concentrations at Mount Hehuan. Finally, the net influence of BB
activities on airborne As concentrations has been simply estimated by comparing the
differences of As concentrations between BB and non-BB days. The result showed, on
average, the contribution of BB activities over S Asia to airborne As was
approximately 1.0 ng m$^{-3}$, which accounted 63% for total airborne As concentrations
in the springtime. Moreover, a ratio of $\Delta As/\Delta CO$ ($\sim 0.00001$) in the S Asian BB events
was obtained. Using this value, arsenic emissions from S Asian BB activities were
estimated to be 0.17 tons yr$^{-1}$, causing extremely high airborne As concentrations over
the subtropical free troposphere, and impacted As cycles on a regional scale.

Key words: Arsenic; Subtropical free troposphere; South Asia; Biomass burning;

As/Pb ratios.


**1.  Introduction**

Arsenic (As), categorized into carcinogenic species by International Agency for

Research on Cancer, is a toxic element and even in trace concentration may exert
hazard to human health. It is also the most highly accumulated trace metal in the
human food chain. Consequently, As has been an environmental concern in terms of
its emissions, cycling and health effects (Nriagu, 1989; Bissen and Frimmel, 2003;
Wai et al., 2016). Atmospheric As is released from both natural and anthropogenic
sources with a total annually global emission of nearly 31 Gg (Nriagu, 1989; Wai et



al., 2016; Walsh et al., 1979). The quantity of As emissions derived from
anthropogenic sources is about 1.6 times higher than that of natural origins (Nriagu,
1989). Arsenic released from volcano is the predominant source of natural emissions,
followed by wind-erosion soil particles as well as biogenic emissions (Nriagu, 1989).
For anthropogenic sources, metal smelting and coal combustion release quantities of
arsenic into atmosphere (Brimblecombe, 1979; Mandal and Suzuki, 2002), and
thereby are considered to be major origins for airborne arsenic. Besides, biomass
burning (BB) for waste timber treated by As-contained insecticides and crops
contaminated by pesticide might enhance the emissions of airborne particulate arsenic
(Huang et al., 2012; Niyobuhungiro and Blottnitz, 2013). However, whether
BB-derived As can be traveled to long distance and influenced As cycles at its
downstream regions is still an open question.

Biomass burning activity emits large amounts of air pollutants into atmosphere

(e.g. carbon monoxide (CO), carbon dioxide ($CO_2$), nitrogen oxides (NOx), volatile
organic carbon (VOC) and particulate matters (PM))(Streets et al., 2003;Tang et al.,
2003). It impacts not only on local but also on regional air quality, atmospheric
chemistry, biogeochemical process and hydrological cycle along with climate
(Crutzen and Andreae, 1990; Ramanathan, 2001; Pochanart et al., 2003; Tang et al.,
2003; Kondo et al., 2003). Southeast (SE) and South (S) Asia are active biomass
burning regions in the world and BB activities in these continents are mostly caused
by deforestation and agricultural activities. Indonesia, India, Myanmar and Cambodia
are major countries of BB activities (Chang and Song, 2010). Among these burned
areas, BB activities in India are mainly caused by burning of crop residues (~61% of
total burning) and frequently occur from January to May and usually maximizes in
springtime (Nriagu, 1989; Pochanart et al., 2003). After burning, a large quantity of



air pollutants with BB plumes would uplift from ground level to free troposphere (2-6
km), transporting to the Pacific region by prevailing westerly wind, and then impact
on the properties of atmospheric chemistry in the downwind regions (Kondo et al.,
2003; Lin et al., 2009).

Over the past decade, numerous studies have shown that west Bengal of India

and Bangladesh are extremely As-contaminated areas in South Asia (Robert et al.,
2010; Neumann et al., 2010; Burgess et al., 2010). The extremely As-contaminated
ground water in these areas is used for both drinking and irrigation. Thus,
accumulation of As would be found in rice roots and rice plants along with crop soils
(Norra et al., 2005). While burning As-contaminated plants, As would be expected to
attach within BB-originated aerosols and probably condense on the existed aerosols,
and transport to the downwind site, enhancing the atmospheric As concentrations in
aerosol phase.

Mountain-top site, which is generally situated far away from direct influence of

local anthropogenic emissions, is very sparsely in the Northern Hemisphere. Due to
the high elevation, mountain-top site is useful to monitor long-range transported air
pollutions (Weiss-Penizas et al., 2007; Lin et al., 2013). From September 2011 to
September 2012, the continuous measurements of total suspended particulate (TSP,
dynamic diameter less than 100 μm), ozone and carbon monoxide were carried out at
Mountain Hehuan in Taiwan, with the aim to better understand the behaviors of air
pollutants transported horizontally from Asian continent and intruded vertically from
high-troposphere/low-stratosphere over the subtropical region. Chemical
compositions of TSP samples, including water-soluble ions and elements, were
analyzed. In this paper, we present the As concentrations and its seasonality at Mount
Hehuan. The potentially regional sources of high As concentrations are also examined



by backward trajectory analyses and WRF-Chem model simulations. Finally, the net
influence of SE and S Asian BB activities on airborne As over the subtropical free
troposphere is assessed. To our best knowledge, this is the first paper to report
regionally transported arsenic accompanying with BB plumes and enhancements in
airborne As concentrations over the subtropical free troposphere.

**2.  Method**
*2.1 Aerosol sampling*

Daily TSP samples were collected at Mount Hehuan site (24.16°N, 121.29°E,

3001 m a.s.l., see in Figure 1) from September 2011 to September 2012. The sampling
station is located in a pristine environment and its vicinity is generally higher than
2900 m, and thereby the monitoring site can be considered as representative of the
free troposphere over the subtropical Pacific region (Lin et al., 2013). A high-volume
TSP sampler (TISCH, Model TE-5170D), operated at a flow rate of approximately
1.13 $m^3$ $min^{-1}$, was used to collect aerosol samples. Whatman®41 cellulose filters (8"
× 10") were used as filtration substrates. After sampling, each filter was folded and
stored in a separate plastic bag that was then stored in a polypropylene container,
frozen immediately, and returned to the laboratory for further chemical analysis.
Carbon monoxide, a tracer for tracking anthropogenic plumes, was monitored by a
nondispersive infrared spectrometer (Horiba model APMA-370). The details of the
instrument and QA/QC procedure for CO monitoring are described elsewhere (Lin et
al., 2013).

*2.2 Chemical Analysis*

For the purpose of chemical analyses, the sampled filter was subdivided into





eight equal pieces after sampling. One piece was subjected to acidic digestion for
elemental determination and another one was extracted by Milli-Q water for
analyzing water-soluble ions. For acidic digestion, each filter sample was put into an
acid-cleaned vessel and digested in a mixed acidic solution (4 mL 60% $HNO_3$ + 2 mL
48% HF) by an ultrahigh throughput microwave digestion system (MARSXpress,
CEM Corporation, Matthews, NC, USA). The digestion process was performed in
three steps: (1) heating to 170°C for 8 min and maintaining this temperature for 7 min
at 1440 W, (2) heating to 200°C for 7 min and maintaining this temperature for 15
min at 1600 W, and (3) cooling for 60 min. Subsequently, the vessel was transferred
to XpressVap$^{TM}$ accessory sets (CEM Corporation) for the evaporation of the
remaining acids until nearly dry. Approximately 2 mL concentrated $HNO_3$ was added
into the vessel and reheated. The resulting solution was then diluted with Milli-Q
water to a final volume of 50 mL. After acidic digestion, 31 target elements in TSP
samples were analyzed through inductively coupled plasma mass spectrometry
(ICP-MS; Elan 6100; Perkin Elmer$^{TM}$, USA). A multi-element standard, prepared
from stock (Merk) composed of 2% $HNO_3$ solution, was used for calibration. An
internal standard containing indium (10 ng mL$^{-1}$) was used to correct instrumental
drift. To minimize the isobaric interference, the nebulizer gas flow rate was adjusted
to 0.7 - 0.9 L min$^{-1}$. To reduce formation of doubly charged ions and oxides, $Ba^{++}$/Ba
and CeO/Ce must be lower than the recommended values of 0.01 and 0.02,
respectively. Accuracy and precision were assessed by replicate measurements (N=7)
of the standard reference material NIST SRM 1648, following the total digestion
process. The results showed that the recoveries for most elements fell within 90-110%
and the precisions were less than 5%. For each run, a blank regent and three filter
membrane blanks were subjected to the same procedure as that for the aerosol



samples. The method detection limits (MDLs) were 0.01 ng m$^{-3}$ for both As and Pb.
Another half of the filter sample was extracted with 20 mL Milli-Q water (18.2
Ω) by using ultra-sonic apparatus for 1h. The extracted solution was subsequently
analyzed for water-soluble ions, including $Na^+$, $NH_4^+$, $K^+$, $Mg^{2+}$, $Ca^{2+}$, $Cl^-$, $NO_3^-$, and
$SO_4^{2-}$, by ion chromatography (Dionex ICS-90 for cations and ICS-1500 for anions)
equipped with a conductivity detector (ASRS-ULTRA). A QA/QC program including
calibration, recovery and precision test along with MDLs for all ions was conducted
during the analyzed processes. A multi-ion solution (Merck) was used for calibration
of IC instrument and seven-point calibration curves were made for each batch of
samples. One laboratory blank was taken for each batch analysis and MDL was
calculated as 3 times standard deviation of the values of 7 blanks. The average
recoveries for all species were in the range of 91-105%; the precisions for all species
were less than 5%.

***2.3 Backward trajectory analysis***
To identify potential sources of airborne arsenic at Mount Hehuan, five-day
backward trajectories were computed by the Hybrid Single-Particle Lagrangian
Integrated Trajectory (HYSPLIT) model developed by the USA NOAA Air Resources
Laboratory (Draxler and Hess, 1998). The meteorological data for the trajectory
model was the GDAS (Global Data Assimilation System), which were processed by
the NCEP with a 6-h time resolution, about 190 km horizontal resolution, and 23
vertical levels. In this work, five-day backward trajectories arriving at 3000 m a.s.l.
were computed once every day at 12:00 LT (local time) with a time step of 6 hours.
Four additional trajectories were generated of which starting locations were changed
± 0.5° from the actual sampling site to reduce the uncertainty of the trajectory analysis.


During the sampling period, a total of 1865 backward trajectories were computed, and
five clusters of air parcels, namely, Northern China (NC), Pacific Ocean (PO), South
Sea (SS), Southeast Asia (SEA) and South Asia (SA), were categorized. Figure 1
shows the pathways of five different air clusters at Mount Hehuan. The frequency of
SA was 33%, which was the predominant air clusters, followed by PO (24%), SS
(18%), SEA (18%) and NC (7%). In the NC group, the air mass originated mainly
from Northern China, where heavily polluted air is contaminated by industrial
emissions, moving to the south areas slowly and then arrived at the receptor site. The
NC air cluster was predominately found in March, August and September with a
frequency of >16% (shown in Figure 2). In case of PO, the air parcel generally
originated from Western Pacific Ocean, spending much time in marine atmosphere
before arriving at Taiwan. This air cluster was most predominately found from July to
September with a frequency of > 48%. High frequency (>20%) of PO cluster was also
surprisingly found in October and November. For SS air cluster, the air parcel was
regularly from South Sea, crossing the marine areas or Luzon Islands, and then
arrived at Mount Hehuan. This air group accounted for 18% with a high frequency in
June, July and November. For SEA group, the air mass typically came from
Indo-China Peninsula, occasionally passing across polluted Southern China, like
Chongqing and Pearl River Delta (PRD) region, before reaching Taiwan. The SEA air
group was profoundly occurred from March to June with the frequency exceeding
30%. Finally, the air parcel of the SA cluster was mainly from Middle East and Indian
Subcontinent, passing over northern parts of Myanmar, Thailand, Laos and Vietnam
along with PRD region, and then descended to Mount Hehuan. The SA group was
frequently found during the sampling periods, except for July to September. The air
masses of NC, SEA and SA groups were associated with continental origins as they



spent much time in Asia continent before arriving at Mount Hehuan. The continental
air masses were mostly prevailed from mid-autumn to late spring (see in Figure 2).
On the contrary, PO and SS air clusters were grouped into marine air parcels and were
profoundly found from June to September. Nevertheless, the air parcels from NC,
SEA and SA groups would be anticipated picking up polluted air and transporting to
Mount Hehuan compared with PO and SS air clusters that spent much time in marine
atmosphere.

*2.4 WRF-Chem model*
The WRF model coupled with chemistry module (WRF-Chem; Ver. 3.2.1) was
employed to study pathways of long-range BB plumes transported to Mount Hehuan
(Skamarock et al., 2008; Grell et al., 2005). WRF-Chem model has been successfully
simulated behaviors of BB plumes transported to the subtropical free troposphere (Lin
et al., 2014). The meteorological initial and boundary conditions for WRF-Chem were
acquired from NCEP-FNL Global Forecast System (GFS) 0.5°×0.5° analysis data sets
(35 vertical levels). The Mellor Yamada Janijc (MYJ) planetary boundary layer
scheme was selected in this study. The horizontal resolution for our BB simulations
was 27 km. To assure the meteorological fields were well simulated, the
four-dimensional data assimilation (FDDA) scheme was activated based on the
NCEP-GFS analysis data.

**3.  Results and discussion**
*3.1 Overview of Airborne Particulate As*
A total of 302 daily TSP samples were collected at Mount Hehuan during the
sampling period. Each TSP sample has been determined the concentrations of water-



soluble ions and elements by IC and ICP-MS, respectively. Because the net mass of
each collected aerosol sample was not measured, the abundance of each species
relevant to TSP mass cannot not be obtained. Figure 3 displays the average
concentrations of ionic species together with metallic elements in TSP samples.
Without determination of carbon contents, sulfate was the most predominant species
in airborne TSP samples with a mean concentration of 4.1 μg m$^{-3}$, followed by nitrate
(2.0 μg m$^{-3}$), ammonium (1.7 μg m$^{-3}$) and chloride (0.23 μg m$^{-3}$). Aluminum (Al), a
typical geological material, exhibited a mean concentration of 184 ng m$^{-3}$, which was
the predominant elements. In addition to K, Ca and Fe (up to 100 ng m$^{-3}$) were also
major metals, followed by Na, Mg, Cu, Ti, Zn and P (10 to 100 ng m$^{-3}$), and then
followed by Pb, Mn, Ba and Sr (1 to 10 ng m$^{-3}$). The rest metals had concentrations of
< 1 ng m$^{-3}$ over the free troposphere.

Arsenic, a target element in this study, exhibited a daily concentration from 0.02

to 5.9 ng m$^{-3}$ with a mean value of 0.5±1.0 ng m$^{-3}$ (Figure S1). As expected, arsenic
concentrations in the continental air groups, such as SA, NC and SEA, were much
higher than those in the marine air categories (Figure 1). The As concentrations (~0.1
ng m$^{-3}$) in PO and SS air groups were in agreement with that of Mauna Loa, Hawaii
(Zieman et al., 1995), indicating that the low As value can be considered as a
background concentration in the subtropical free troposphere (Zieman et al., 1995). A
large standard deviation suggested that As concentration at this mountainous site had
a large day-to-day variation. Increased As concentrations coincided with CO peaks on
some days, showing some highly anthropogenic As plumes passed over this site.
Some As peaks were found with enhancements of both CO and potassium ion (K$^+$),
especially in the springtime, indicating BB origins.

Figure 4a shows monthly variations of 25th, 50th and 75th percentile values of



arsenic concentrations observed at Mount Hehuan. As can be seen, the median
concentration of arsenic increased from January (0.18 ng m$^{-3}$), maximizing in May
(0.81 ng m$^{-3}$), and then decreased abruptly through June to December (from 0.05 ng
m$^{-3}$ in June to 0.13 ng m$^{-3}$ in August). The seasonality of As was different from those
of Al (a tracer of dust) and K$^+$ (a marker of BB) as shown in Figures 4b and 4c, but
was very similar to that of Pb (Figure 4d), suggesting As and Pb might be originated
from the similar sources. The seasonal distributions of As at this mountainous site
were associated with emission sources, regional circulations and local meteorological
conditions. Marine air prevailed from July to November, except October, resulting in
lower As concentrations over the subtropical free troposphere. On the contrary,
continental air prevailed in the wintertime and springtime, picking up polluted air and
transported to the receptor site; as a result, increase of As concentrations was expected.
Besides, favorable locally meteorological conditions for dispersion of air pollution
might be another reason for the lower As concentrations in the summertime (Lin et al.,
2011; 2013). The 5th percentile value for As concentration is better to understand the
distributions of extremely high As events over the free troposphere. Higher 5th
percentile values of arsenic were found between February (0.99 ng m$^{-3}$) and May
(1.27 ng m$^{-3}$) compared to those of other seasons (from 0.09 ng m$^{-3}$ in November to
0.60 ng m$^{-3}$ in September), reflecting more high-As plumes crossed over Mount
Hehuan from late-winter to late-spring. Over the subtropical free troposphere, two
distinct haze plumes were usually observed from late winter to spring: one is dust
storm that originated from East-Asian and non-East Asian continents (Lin et al., 2001;
Hsu et al., 2012); another one is BB plume which mainly comes from SE and S Asia
(Lin et al., 2009; 2010). Both plumes would impact the atmospheric compositions, of
course, including airborne As in Pacific region. As shown in Figures 4b and 4c,



substantially elevated Al and $K^+$ concentrations were observed in the springtime,
especially for 25th percentile values, suggesting that Mount Hehuan was influence by
both dust and BB aerosols.

Enrichment factor (EF) analysis referred to average crust (Taylor, 1964) can be

differentiated natural from anthropogenic sources for particulate elements. The EF
value for each element in TSP sample $(EF(X_i)_{TSP})$ can be calculated as:

$$EF(X_i)_{TSP} = \frac{(X_i/Al)_{TSP}}{(X_i/Al)_{Crust}} \qquad (1)$$


where $(X_i/Al)_{TSP}$ is the concentration ratio of a given element X to Al in airborne TSP
samples and $(X_i/Al)_{Crust}$ is the concentration ratio of an interested element X to Al in
the average crust (Taylor et al., 1964).

Figure 5 plots the EF values for analyzed elements in different seasons. The EF

values for Fe, Ti, Mg, Sr, Mn, Ca and Rb along with some rare earth elements, such as
Nd, Ce and La, were normally less than 5 in various seasons (the annual average EF
values ranged from 1.0 for Fe to 3.6 for Rb), indicating crustal origins. On the
contrary, the EF values for Tl, Mo, Sn, Zn, Pb, Cu, Sb, Cd and Se were regularly
higher than 10, suggesting anthropogenic sources. Of course, high EF values were
found for arsenic in all seasons (ranging from 114 in the autumn to 359 in the winter),
suggesting that particulate As in the free troposphere was mainly from anthropogenic
emissions, but not from natural wind-erosion soil throughout the whole sampling
period; nevertheless, biomass burning is a candidate for high As source in the
springtime, which will be further discussed in section 3.2.

*3.2 Potential source for As in the BB seasons*

Figure 6 shows the scattered plots of As against $K^+$, Al and Pb in different



arsenic concentration bins. We found that As correlated well with $K^+$ ($r = 0.78$, $p < .05$
for the 5th percentile value of As) when severely high As events occurred, suggesting
BB origins. Oppositely, arsenic correlated poorly with Al ($r$ ranged from 0.05 to 0.42)
in all As concentration bins, indicating that wind-erosion soil was not a major source
for airborne As at the sampling site. However, significantly positive correlations were
observed between As and Pb within 25th percentile values of As concentrations,
reflecting that airborne As and Pb were from the same sources in the high arsenic
events.

As mentioned above, BB activities may be an important regionally source for

high As concentrations over the subtropical free troposphere, especially during the
spring period; consequently, in this section, we prove the hypothesis using backward
trajectory analyses and MODIS fires observations together with WRF-Chem model
simulated results. Figure S2 shows the seasonality of fire spots over SE and S Asia
observed by MODIS from 2011 September to 2012 September. In SE Asia, the BB
activities showed strong seasonal variations with a gradual increase from January to
March, when it reached a peak. It then decreased substantially from late spring to a
minimum in summer. In South Asia, the total annual counts of fire spots were
approximately 20% of that in SE Asia. Similar seasonality was found with intensive
fire spots in the springtime and maximum in May. The fire spots then decreased
during summer to mid-winter and minimized in July. However, the total fire spots (SE
Asia plus S Asia) maximized in March. This might explain why particulate $K^+$ and
CO concentrations at Mount Hehuan were highest in March.

For convenience, prior to further analysis we arbitrarily chose a $K^+$ concentration

of 109 ng m$^{-3}$ (the 25th percentile value of potassium ion) as a criterion value for
identifying the suspected BB event. A second criterion (CO concentration up to 160





ppb) was also added for selection of the BB plume. Ultimately, a total of forty-eight
suspected BB TSP samples were identified during the entirely sampling period. Figure
7 shows time series of daily concentrations of As, $K^+$ and CO observed at Mount
Hehuan from January to May, 2012 when intensive BB activities were occurred over
SE and S Asia. The air clusters are also shown in this figure for helping to identify the
air origins. As can be seen, several As spikes coincided with increasing CO and $K^+$
(e.g. Feb. 19, Mar. 31, Apr. 3, May 5 and 7 etc.), implying BB origins. Backward
trajectory showed that the air parcels for the high arsenic events originated mainly
from SA air group and passed over fire regions. A high arsenic plume passed over
Mount Hehuan with As concentration increasing from 1.2 ng m$^{-3}$ on 25 March to 5.3
ng m$^{-3}$ on 3 April though low As concentration was found on 2 April. Figure S3a
illustrates the five-day backward trajectories starting at Mount Hehuan during this
period. The result showed the air parcels mainly passed over northern India, Nepal,
Bangladesh and Southeast China before arriving at Taiwan. Figure 8a plots the
distributions of MODIS fires from March 25 to April 3, and WRF-Chem model result
at an altitude of 700 hPa on April 3 when the high daily As concentration (5.3 ng m$^{-3}$)
was observed. In this case, extensive fire spots were observed over northern part of
India from March 29 to April 2; the BB plume originated over burned areas,
transporting to east direction, and passed over Mount Hehuan, resulting in increased
concentrations not only for $K^+$ and CO, but also for arsenic. As shown in Figure 9a,
during the BB events over the S Asian continent, arsenic correlated well with $K^+$ ($r$
=0.73, $p < .05$). On the contrary, the correlation coefficient between As and $K^+$ in the
non-BB events was 0.53 ($p > .05$). This supported our argument, that is, airborne
arsenic at Mount Hehuan was attributed to BB activities over S Asia. However, some
BB plumes were observed at Mount Hehuan, but the As concentrations were not





elevated. For example, a suspected BB plume was found from March 8 to 14 since $K^+$
and CO concentrations increased concurrently. Based on backward trajectory analysis,
the air parcels during this BB event were mainly from SE Asia, passing over southeast
China, and then arrived at Mount Hehuan (Figure S3b). Intensive fire spots observed
in Indo-China Peninsula and WRF-Chem modelling jointly confirmed that the BB
plume also across Taiwan (Figure 8b). Nonetheless, the As did not rise, but kept at the
low levels of 0.2 ng $m^{-3}$. Another similar case was also found in the end of February
(Feb. 26 to 29). Unlike BB events over S Asia, arsenic correlated weakly with $K^+$ ($r =$
0.4, p >.05, Figure 9b) in the BB events from SE Asia, as well as that in the maritime
air groups (Figure 9c). These findings suggested that some specific sources might
release numerous arsenic into atmosphere during BB activities over S Asia, but not
over Indo-China Peninsula. Wind-erosion soil particles are one of important sources
for airborne arsenic. According to the investigation by Nriagu (1989), arsenic derived
from wind-erosion dust was 2.1 Gg $yr^{-1}$, accounting 18% for natural As emissions.
Figures S4a – S4c show the scattered plots of As against Al in all air groups during
the S and SE Asian BB periods. Poor correlations were found between As and Al in
the various air groups, except for the SS air category ($r = 0.88$, $p <.05$), indicating that
wind-erosion soil was not a major source for As over the free troposphere.
Interestingly, a good correlation of As and Al was found in the SS air group. The
marine air parcels, which spent a long time in the clean marine atmosphere, are
subjected to dilution which can affect the air pollution (Lin et al., 2011), probably
resulting in similar behaviors of As and Al.

Recently, numerous studies pointed out S Asia, especially in west Bengal of

India and Bangladesh, are extremely As-contaminated areas (Burgess et al., 2010;
Neumann et al., 2010; Roberts et al., 2010;). In these regions, highly As-contaminated



ground water, typically caused by geological process, is not only used for drinking
water, but is also used for irrigation of crops. Accumulation of arsenic has been found
in rice roots and rice plants along with crop soils (Norra et al., 2005). After burning,
the As might be released from these crops into atmosphere, and transported easterly to
Pacific regions with BB plumes. On the other hand, uses of pesticide as an insecticide
for cotton, paddy and wheat in India and Bangladesh might be another reason for As
contamination in crops (Aktar et al., 2009). Lead arsenate (LA, $[Pb_5OH(AsO_4)_3]$;
As/Pb~0.22) was the most extensively used of the arsenical insecticides in the world.
Although LA was officially banned as insecticide in 1990's in many developed
countries, but has not been banned in India nowadays. Figures 10a and 10b shows the
scattered plots of As against Pb in TSP samples for various air groups during the BB
season. The higher As concentrations were generally found in the SA air category. In
case of SA air group, the average As concentration in the BB events were 1.6±1.4 ng
$m^{-3}$, exceeding that (0.6±0.7 ng $m^{-3}$) in non-BB events by a factor of 2.7 ($p < .05$),
suggesting a special arsenic emission source over S Asian continent during the BB
season. In some cases, low As concentrations were also found when the BB plumes
transported from S Asia. The reason has not been clearly understood, but might be
explained by a mixed source of the BB plume with other emissions during the air
transportation. In terms of SEA group, no substantial discrepancy of As
concentrations was found during BB and non-BB periods, indicating that BB over
Indo-China Peninsula was unable to enhance As concentrations over the subtropical
free troposphere.
During the S and S Asian BB period, good correlations between As and Pb
(ranging from 0.84 for SA-BB to 0.96 for NC, see in Figure 10) were found in various
air groups; hence, a ratio of As/Pb might be given us an insight to trace the



specifically regional arsenic emissions in SA air group when BB activity occurred.
During the SA-BB plumes, the average As/Pb ratio was 0.18 (see in Figure 10a),
which was much higher than the average value (0.11) of non-BB (SA-non-BB) events
along with those (ranging from 0.08 to 0.1) of other air categories (see in Figures 10b
and10c), implying a special source for As during the BB events over S Asia. Some
data sets of SA-BB groups showed low As/Pb ratios, probably reflecting mixed air of
BB plumes and other emission sources transported to the subtropical free troposphere.
Wind-erosion soil particles and metal smelting (lead smelting) along with coal
combustion industries are major natural and anthropogenic sources of airborne As,
respectively. In Northern India, As/Pb ratio in natural soil, paved road and unpaved
road dust varied from 0.02 to 0.13 while low As/Pb ratios were found in lead smelting
(0.002), coal combustion in stoves (0.0016) and coal fire power plants (0.0026) (Patil
et al., 2013). Our As/Pb ratios in the SA-BB events were much higher, suggesting that
wind-erosion dust, lead smelting and coal combustion seemed not to be major sources.
In particular, the As/Pb ratio was normally higher than 0.20 when severely high As
concentrations were observed. This ratio was in line with that of LA (~0.22),
suggesting that burning crops contaminated by LA in S Asia could be a crucial
candidate for extremely high As concentrations at Mount Hehuan during the BB
periods.

**3.1  Impact of Biomass Burning**

The difference of As concentrations between the BB and non-BB days could be

considered as the net influence of BB activities on airborne As concentrations over the
subtropical free troposphere (Lin et al., 2010; 2013). Table 1 lists the differences of As,
Pb, $K^+$ and CO concentrations of BB and non-BB samples in SA and SEA air groups.



For SA air cluster, all species increased apparently in the BB events. On average, the
As concentrations during the BB and non-BB periods were 1.6 and 0.6 ng m$^{-3}$,
respectively. The difference of As concentrations between the BB and non-BB events
was 0.6 ng m$^{-3}$. On the contrary, the differences of concentrations in K$^+$ and CO were
observed in the BB and non-BB events for SEA air clusters, but not found for As and
Pb. Again, this suggested that BB activities from SE Asia would not release enormous
arsenic into atmosphere and transport to the subtropical free troposphere by westerly
belt. Assuming that net difference of As concentrations in the BB and non-BB events
was mainly contributed by BB activities, we then obtained that BB activity over S
Asia contributed approximately 63% of airborne As in the subtropical free
troposphere during the BB seasons.

As listed in Table 1, the BB air masses emitted from the S and SE Asian

continents contained $\Delta K^+/\Delta CO$ ratios of 0.0043 and 0.0018, respectively. Each value
was in the same order of magnitude of that estimated by Tang et al. (2003) who
claimed the BB events emitted from SE Asia had a $\Delta K^+/\Delta CO$ ratio of 0.0038. Besides,
a ratio of $\Delta As/\Delta CO$ in the S Asian BB events was estimated to be 0.00001, which was
one order of magnitude higher than that ($\Delta As/\Delta CO\sim0.000001$) of SE Asian BB events,
indicating that much more As released into atmosphere from the S Asian continent.
According to the emission inventory, the annual CO emission rate from biomass
burning over S Asia was nearly 17 Gg y$^{r-1}$ (Stress et al., 2003), we then roughly
estimated that approximately 0.17 tons yr$^{-1}$ of arsenic was released into atmosphere
due to S Asian BB activities, resulting in enhancements of As concentrations over the
subtropical free troposphere in the springtime.

**4.  Conclusion**



Daily TSP samples were collected at Mount Hehuan from September 2011 to
September 2012, in order to investigate the behaviors of long-range transported
particulate matters and their impact on atmospheric chemistry over the subtropical
free troposphere. Arsenic, a target metal in TSP samples, were determined by ICP-MS.
The results showed the daily As concentrations varied from 0.02 to 5.9 ng m$^{-3}$ with a
mean value of 0.5±1.0 ng m$^{-3}$. Some extremely high As concentrations coincided with
concurrent enhancements of K$^+$ and CO, indicating BB origins. Backward trajectory
and WRF-Chem model proved that the high As plumes originated mainly from S Asia.
The ratio of As/Pb (>0.2) in high As events elucidated burning crops contaminated by
lead arsenate might be an important source at Mount Hehuan in the springtime.
Furthermore, we roughly estimated that approximately 1.0 ng m$^{-3}$ of As was
contributed by biomass burning activities over the South Asian continent, accounting
63% of total airborne As in the springtime. Biomass burning over S Asia produced a
As/CO ratio of 0.00001 and released approximately 0.17 tons of As into atmosphere
every year, causing increase in As concentrations over the subtropical free
troposphere.
Asian continent is well known a big source of airborne As in North Pacific
region. Previously, high As concentrations over free troposphere in Northern Pacific
region have been considered as contributions of industrial emissions (Perry et al.,
1990; Wai et al., 2016). From our study, we proposed a new concept for a potential
source of high As over the subtropical free troposphere, that is, BB activities over S
Asia might be an important source of airborne arsenic. In this study, arsenic emissions
from S Asian BB activities was estimated to be 0.17 ton yr$^{-1}$. Compared to the
globally anthropogenic arsenic emissions (~18.8 Gg yr$^{-1}$, Nriagu and Pacyan, 1988),
arsenic released from the S Asian BB activities seemed to be neglected. Indeed, it



contributed a large quantity (~63%) of airborne As over the subtropical free
troposphere in the springtime. Consequently, we concluded that BB activities over S
Asia could certainly impact arsenic cycles on a regional scale that has never been
considered in previous studies.

**Acknowledgements**
This study was financially supported by the Natural Scientific Foundation of
China (No. 91643109), the National Key Research and Development Program of
China (No. 2017YFC0210101), and the Ministry of Science and Technology of R.O.C.
(No. MOST 104-2111-M-001-009-MY2).

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



**Table Captions**

Table 1 The max, min, mean, standard deviation values of As, Pb, $K^+$ and CO of on

the BB and Non-BB days in RA and SA air clusters.

**Figure Captions**

Figure 1 Clusters of backward trajectory at Mount Hehuan from September 2011 to

September 2012.

Figure 2 Monthly distributions of the fractions for various air clusters at Mount

Hehuan during the sampling period.

Figure 3 Average concentrations of chemical compositions in TSP samples collected

at Mount Hehuan site from September 2011 to September 2012.

Figure 4 Monthly distributions of 5th, 25th, 50th, 75th and 95th percentile values of

As concentrations observed at Mount Hehuan from 2011 September to 2012

September.

Figure 5 Average enrichment factors (EFs) of all elements in different seasons.

Figure 6 Scattered plots of As against (a) $K^+$, (b) Al and (c) Pb in different As

concentration bins observed at Mount Hehuan

Figure 7 Time series of daily airborne particulate As, Pb and $K^+$ along with CO

concentrations and clusters of trajectory observed from January to May in

2012.

Figure 8 MODIS fires and WRF-Chem modeled results of BB plumes on (a) April 3

and (b) March 25.

Figure 9 Scattered plots of As against $K^+$ observed at Mount Hehuan in (a) SA, (b)

SEA and (c) other air groups during the S Asian biomass burning seasons.





Figure 10 Scattered plots of As against Pb observed at Mount Hehuan in (a) SA, (b)

SEA and (c) other air groups during the S Asian biomass burning seasons.





Table 1 The max, min, mean, standard deviation values of As, Pb, $K^+$ and CO of on
the BB and Non-BB days in RA and SA air clusters.

| Categories | As (ng m⁻³) | Pb (ng m⁻³) | $K^+$ (ng m⁻³) | CO (ppb) |
|---|---|---|---|---|
| ***SA air cluster*** | | | | |
| Non-BB | | | | |
| Max | 3.5 | 16.9 | 831 | 432 |
| Min | 0.05 | 0.6 | 15 | 102 |
| Mean | 0.6 | 4.5 | 207 | 188 |
| Std. | 0.7 | 3.8 | 173 | 86 |
| | | | | |
| BB | | | | |
| Max | 5.3 | 28.5 | 1617 | 316 |
| Min | 0.13 | 1.6 | 71 | 156 |
| Mean | 1.6 | 10.2 | 404 | 217 |
| Std. | 1.4 | 7.3 | 336 | 42 |
| | | | | |
| Differences[1] | 1.0 | 5.7 | 197 | 29 |
| | | | | |
| ***SEA air cluster*** | | | | |
| Non-BB | | | | |
| Max | 2.3 | 11.0 | 609 | 259 |
| Min | 0.08 | 1.1 | 139 | 170 |
| Mean | 0.6 | 4.2 | 328 | 212 |
| Std. | 0.7 | 3.2 | 178 | 39 |
| | | | | |
| BB | | | | |
| Max | 1.6 | 10.0 | 452 | 282 |
| Min | 0.02 | 0.3 | 4 | 95 |
| Mean | 0.4 | 2.9 | 151 | 148 |
| Std. | 0.4 | 2.5 | 141 | 45 |
| | | | | |
| Differences | 0.2 | 1.3 | 177 | 64 |

1. Difference for each species are calculated by the mean values in BB and
non-BB events.





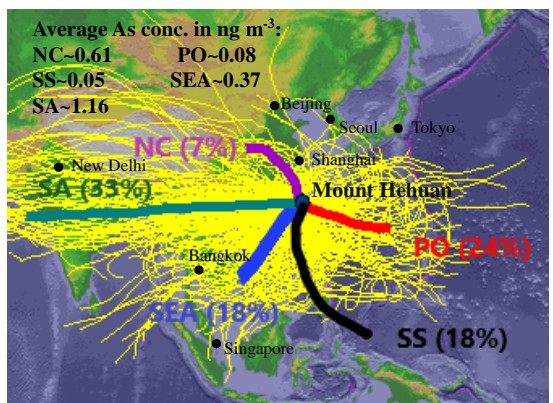

**Figure 1**





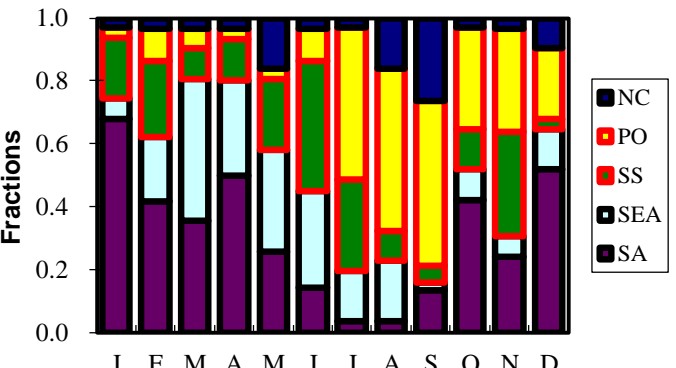

**Figure 2**





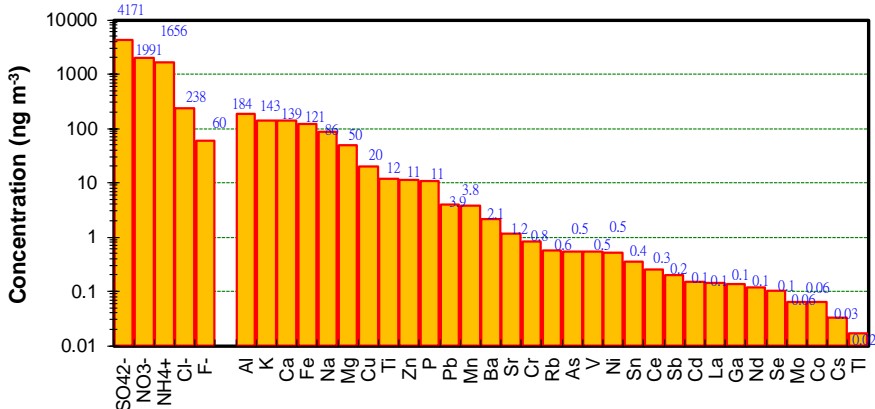

**Figure 3**





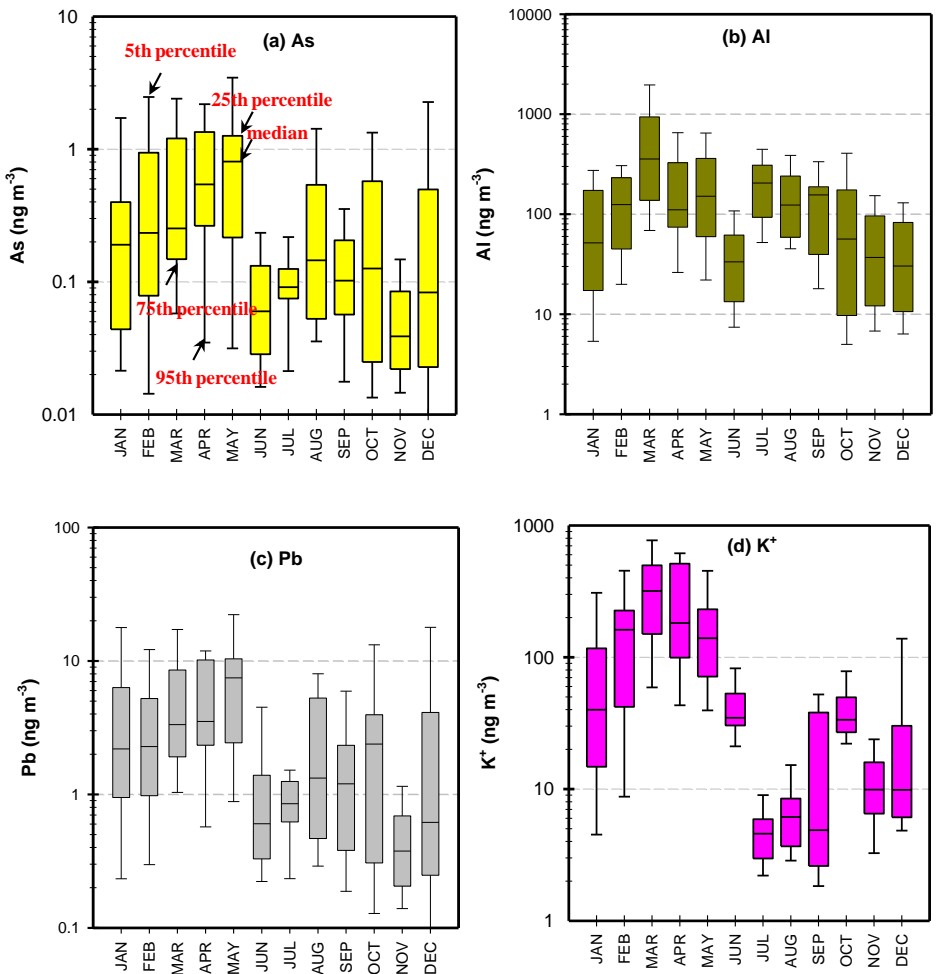

**Figure 4**




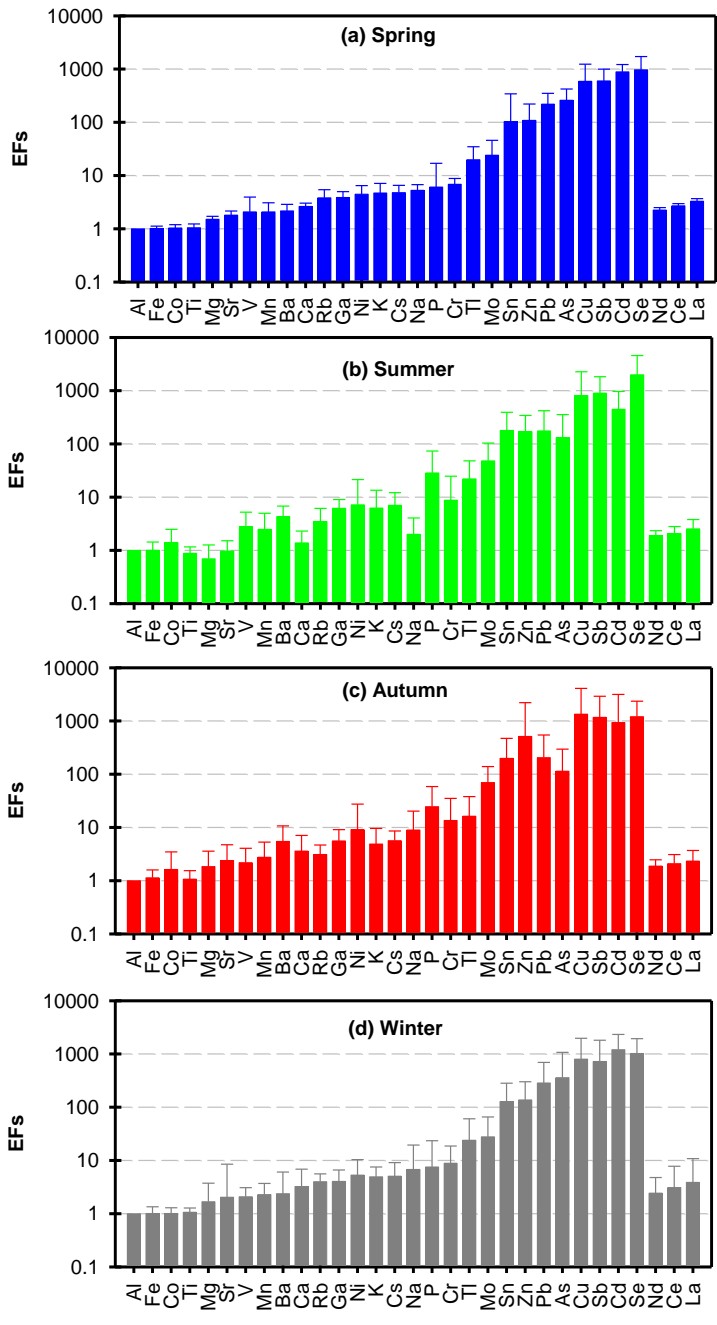

**Figure 5**





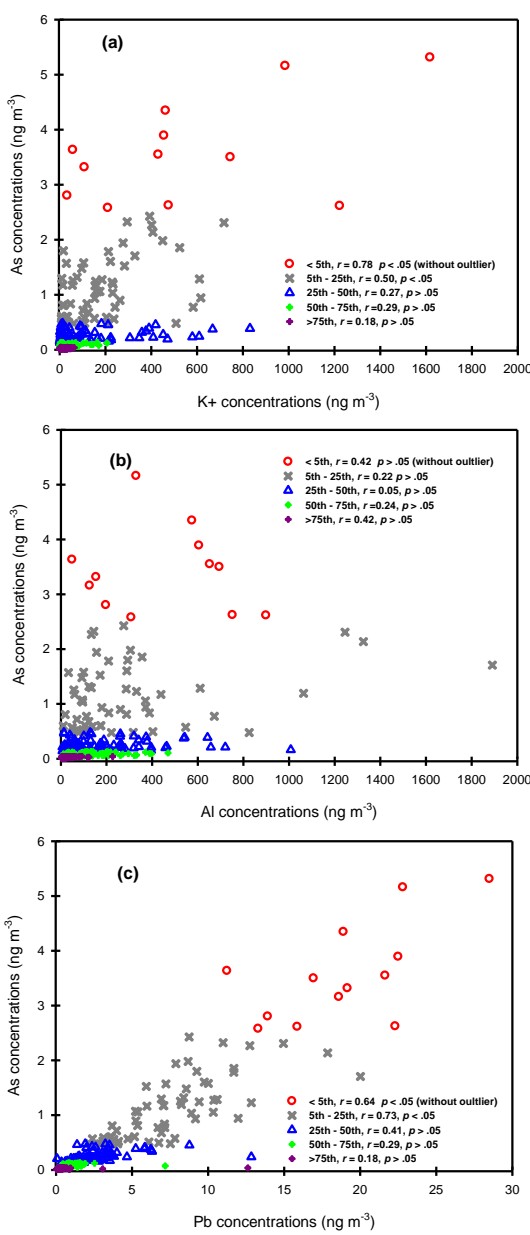

**Figure 6**




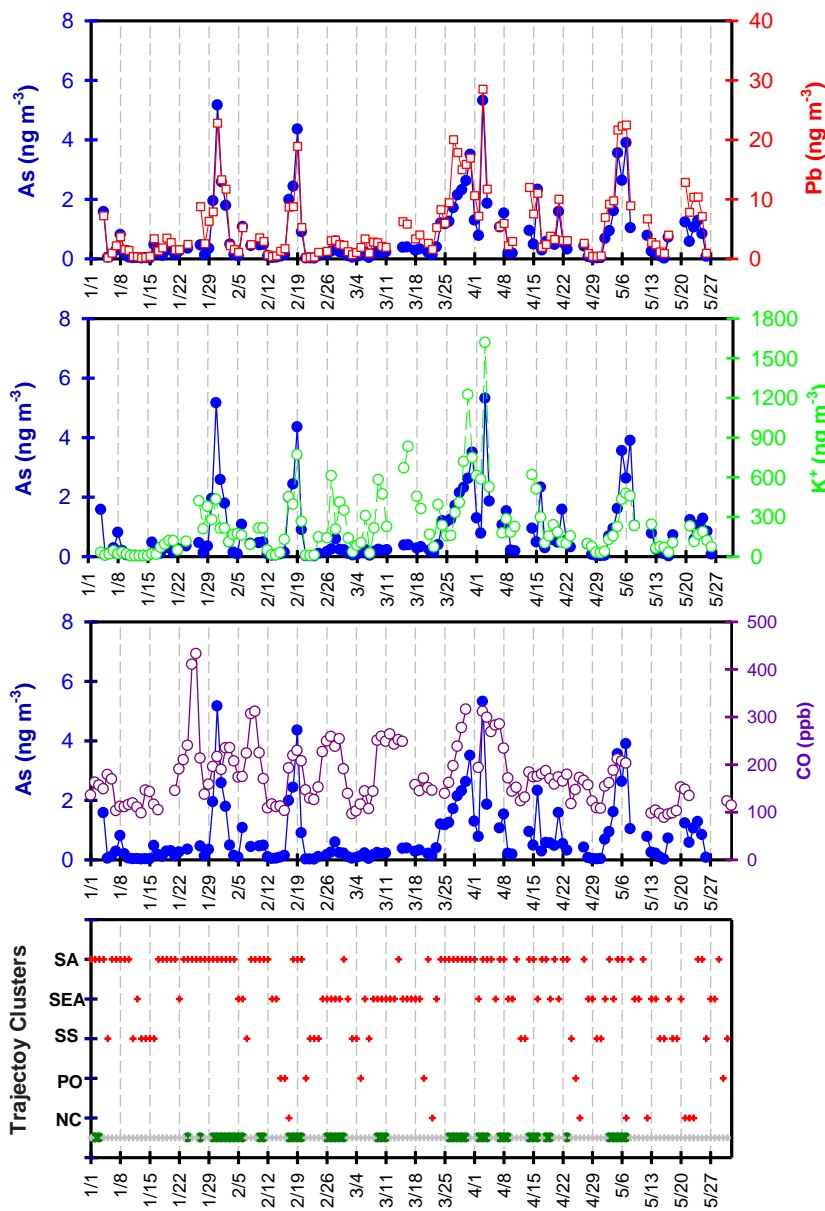

**Figure 7**



**(a)**

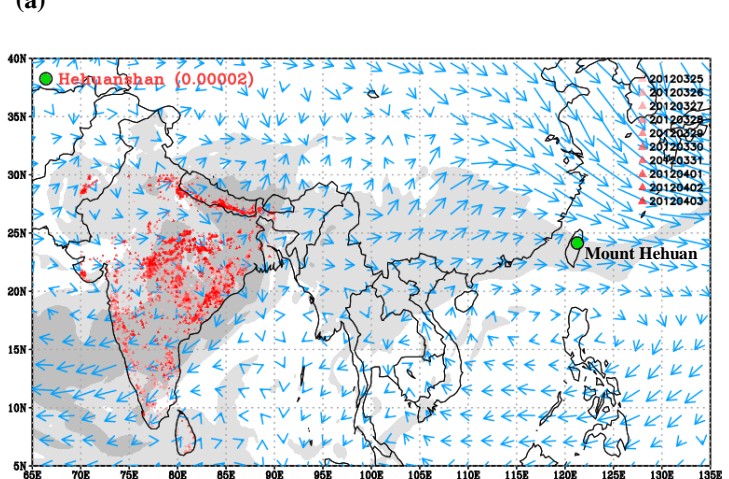

**(b)**

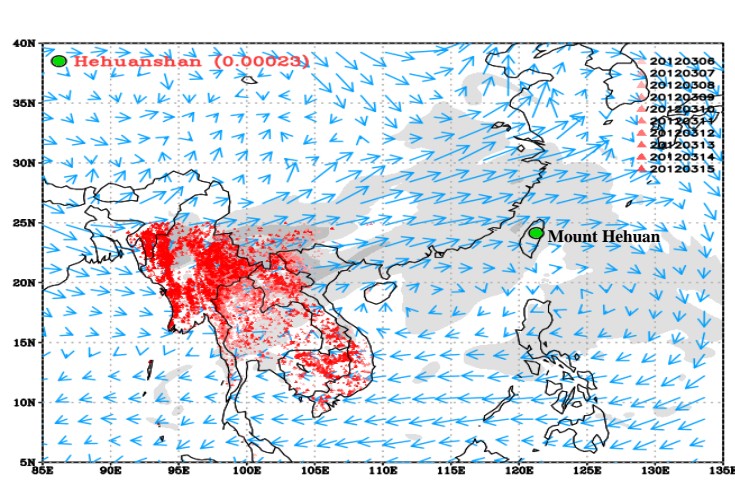

Figure 8



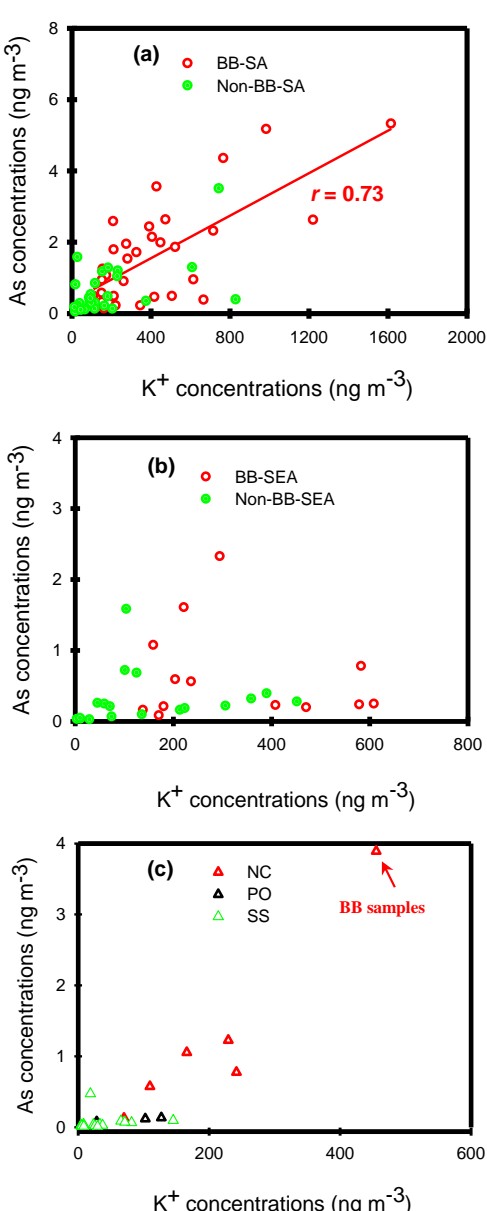

**Figure 9**





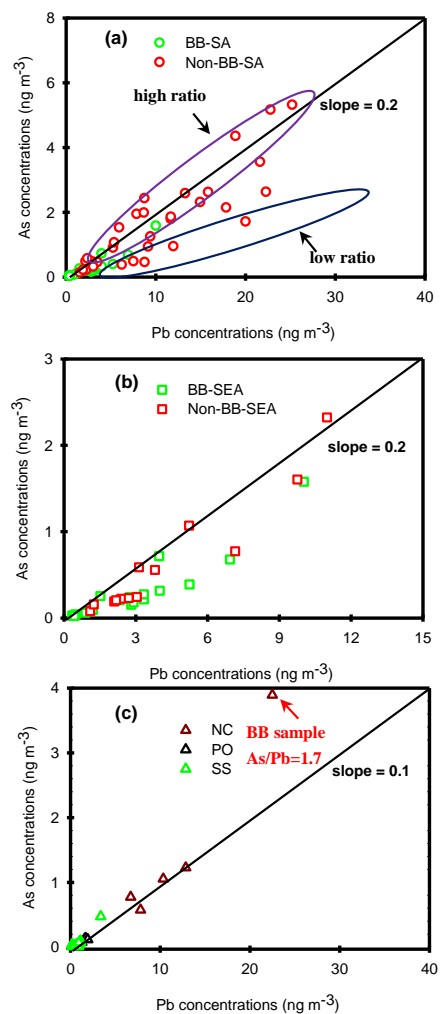

**Figure 10**