# Peer review of "Changes are marked by red color"

_Atmospheric Chemistry and Physics, 2018_

## Referee Comment (RC1) · Anonymous Referee #2 · 10 May 2018

This paper presents one year of daily TSP samples at Mount Hehuan, Taiwan. The samples were analysed for inorganic ions and trace metals and the analysis is supplemented by CO concentration measurements and air mass history analysis.

The manuscript concentrates on the observation of elevated As concentration in biomass burning (BB) plumes. As is found to correlate strongly with Pb, and based on As/Pb ratio of approx. 0.2 the authors suggest that the elevated As concentrations in BB plumes may originate in the usage of lead arsenate pesticide.

While the observation of elevated As in BB plumes is interesting, the concentrations are well below air quality limits at Mount Hehuan. Also the estimated As emission of

0.17 tons per year in South Asian BB is very small (approx. 0.5%) compared to the annual global emissions of 31 tons. Therefore, the results are of only limited interest.

In my opinion this data set requires a more thorough statistical analysis to merit publication in ACP. Please apply principal component analysis (PCA), or a similar factorisation technique, to the data to see if other sources than biomass burning can be identified. Please provide also average concentrations of trace metals and inorganic ions for the different air mass origins you defined. I believe this would provide a valuable reference point for East Asian free troposphere.

Minor comments

L30-35 "Finally, the net influence of BB activities on airborne As concentrations has been simply estimated by comparing the differences of As concentrations between BB and non-BB days. The result showed, on average, the contribution of BB activities over S Asia to airborne As was approximately 1.0 ng m-3, which accounted 63% for total airborne As concentrations in the springtime."

Do you mean that As concentration was on average 1.0 ng m-3 higher during BB days than non-BB days in the springtime? This is quite a different conclusion than "the contribution of BB activities over S Asia to airborne As was approximately 1.0 ng m-3". Please define springtime and state the BB contribution to As on annual level.

L36-38 "Using this value, arsenic emissions from S Asian BB activities were estimated to be 0.17 tons yr-1, causing extremely high airborne As concentrations over the sub-tropical free troposphere, and impacted As cycles on a regional scale."

I wouldn't call the concentration "extremely high" as it is below most (if not all) air quality limits. It is high for free troposphere, but much higher average values have been reported from continental boundary layer. Also, 0.17 tons yr-1 is approx. 0.5% of the global annual emissions, so not a huge amount. However, As in BB smoke may pose a serious health risk close to the fire, where the smoke is not yet diluted.

[Figure]

L60-63 "However, whether BB-derived As can be traveled to long distance and influenced As cycles at its downstream regions is still an open question."

Wai et al. (2016) show that As in general can travel long distances globally. Also it is well known that BB smoke can travel long distances. So it is not a surprise that As released in BB can travel long distances.

L64-65 "volatile organic carbon (VOC) and particulate matters (PM))" Do you mean "volatile organic compounds (VOCs) and particulate matter (PM)" ?

L69 Do you mean "Kondo et al., 2004"? Please check all occurrences of this reference.

L75-77 Most BB smoke is emitted within boundary layer (e.g. Val Martin et al., 2010), though some is of course lifted into free troposphere.

L176 Please explain shortly how the clustering was done. Did you take into account the height of the trajectory?

L209 Please explain which parameters/fields from WRF-Chem did you use in this study. Did you obtain emission sensitivity from the model? Did you take deposition (dry and/or wet) into account?

L231 Are concentrations presented under prevailing conditions or in STP or NTP?

L246-247 "Increased As concentrations coincided with CO peaks on some days, showing some highly anthropogenic As plumes passed over this site." Please explain what you mean with "highly anthropogenic As plumes".

L279-286 Please define the enrichment factor in section 2, it belongs to methods.

L287 and Fig. 5. I don't see much difference between the different seasons in Fig. 5. I think that this is one place where principal component analysis (or similar) would be very useful to differentiate between crustal origins and other sources. See e.g. the analysis by Venter et al. (2017) on trace metal concentrations.

L309-313 "As mentioned above, BB activities may be an important regionally source for high As concentrations over the subtropical free troposphere, especially during the spring period; consequently, in this section, we prove the hypothesis using backward trajectory analyses and MODIS fires observations together with WRF-Chem model simulated results."

This is quite a strong statement. Given the uncertainties in trajectory calculations, I think the correlation with K+ is a stronger indicator of BB origin of As. Please consider re-phrasing.

L313 Please state (with appropriate references) which MODIS product you mean by "fire spots". It would be good to include a sub-section in section 2 detailing which methods are used to characterise BB plumes.

L323-326 "For convenience, prior to further analysis we arbitrarily chose a K+ concentration of 109 ng m-3 (the 25th percentile value of potassium ion) as a criterion value for identifying the suspected BB event. A second criterion (CO concentration up to 160 ppb) was also added for selection of the BB plume."

Do you mean that your site is within a BB plume 75% of time? Using 25th percentile value of K+ as threshold sounds unrealistic to me. At the same time using CO threshold of <160ppb sounds very strange. Please explain.

L332-334 Please plot back-trajectories for high-As BB-plumes and low-As BB-plumes on a map separately (e.g. with different colours). Is there a difference in the footprint area? The back-trajectory clusters have substantial overlap.

Here it would be better to define high and low As plumes based on As/CO or As/K ratio. The absolute As concentration in a plume depends strongly on dilution, whereas the above ratios should remain rather constant independent of dilution. Better still to use excess (delta) As, CO and K+ to calculate the ratios.

It would be interesting to see if principal component analysis finds a factor with high As

and Pb contributions. Inspecting the air mass history of such a factor should make it very clear where the As plumes originate.

L373-378 What is the As/Pb ratio in the ground water in this region? Is it significantly different from the ratio of 0.22 for lead arsenate?

L428-429 Is the difference calculated correct here? Also, this difference depends mostly on BB plume dilution.

L433-436 Please try to find BB-plume As factor with PCA. That will give you a more reliable estimate of the BB-contribution to As load at Mount Hehuan.

L438 How do you define delta? Is this excess concentration in the plume? This should be introduced already in section 2.

L438-442 Please indicate the unit of delta As / delta K+ and delta As / delta CO.

L444-445 For an order of magnitude estimate a 15-year old emission inventory is probably ok, but you may wish to use a newer inventory. At least GFED4 (van der Werf et al., 2017) differentiates between agricultural residue burning and other kind of fires.

L457-458 "Backward trajectory and WRF-Chem model proved that the high As plumes originated mainly from S Asia."

The high As plumes may have originated from S Asia, but in my opinion this has not been proved. Especially Fig. 8, which is the only place where WRF-Chem is utilised, is misleading and poorly documented. Please see comments on Fig. 8 below.

L461-463 "Furthermore, we roughly estimated that approximately 1.0 ng m-3 of As was contributed by biomass burning activities over the South Asian continent, accounting 63% of total airborne As in the springtime."

Here "1.0 ng m-3 of As" sounds strange considering that you present only annual mean (0.5 ng m-3) before. Please clarify and state the BB contribution to As on annual level and define "springtime".

Fig. 6 Please explain how the colour coding (As bins) is done. Now it seems to me that "<5th" refers to highest 5 % of values, though usually 5th percentile would include the lowest 5 % of data points.

Fig.8 Caption: "Figure 8 MODIS fires and WRF-Chem modeled results of BB plumes on (a) April 3 and (b) March 25." Please check if panel (b) presents March 25 or March 15. The "fire spot" legend in the plot suggests March 15 to me.

Please indicate what the blue arrows and the shaded areas represent.

My main concern with this figure is that for panel (a) all fire observations outside India are left out. This gives a very misleading picture of the potential sources of the BB plumes for April 3 observations at Mount Hehuan, as during March 25 – April 3 there are much more fires between India and Taiwan than in India (see attached screenshot from FIRMS).

Also Fig. 8b the "fire spots" are drawn only for a small sub-region of the map.

Finally, there are still quite a few language issues, which should be carefully checked. Please also consider splitting some very long paragraphs into shorter ones (e.g. L166-207 is all just one paragraph).

From the supplement: Please provide proper caption for all figures. Please indicate also here the proper details of MODIS "fire spots".

References

Val Martin, M., Logan, J. A., Kahn, R. A., Leung, F.-Y., Nelson, D. L. and Diner, D. J.: Smoke injection heights from fires in North America: analysis of 5 years of satellite observations, Atmos. Chem. Phys., 10(4), 1491–1510, doi:10.5194/acp-10-1491-2010, 2010.

van der Werf, G. R., Randerson, J. T., Giglio, L., van Leeuwen, T. T., Chen, Y., Rogers, B. M., Mu, M., van Marle, M. J. E., Morton, D. C., Collatz, G. J., Yokelson, R. J. and

Kasibhatla, P. S.: Global fire emissions estimates during 1997–2016, Earth Syst. Sci. Data, 9(2), 697–720, doi:10.5194/essd-9-697-2017, 2017.

Venter, A. D., van Zyl, P. G., Beukes, J. P., Josipovic, M., Hendriks, J., Vakkari, V. and Laakso, L.: Atmospheric trace metals measured at a regional background site (Welgegund) in South Africa, Atmos. Chem. Phys., 17(6), 4251–4263, doi:10.5194/acp-17-4251-2017, 2017.

Wai, K.-H., Wu, S., Li, X., Jaffe, D. J., and Perry, K.D.: Global atmospheric transport and source-receptor relationships for arsenic. Environ. Sci. Technol., 50, 3714-3720, doi:10.1021/acs.est.5b05549, 2016.
* * *
[Figure]

[Figure]

**Fig. 1.**

---

## Referee Comment (RC2) · Anonymous Referee #1 · 10 May 2018

In this work, the authors choose Mount. Hehuan (3000 m asl) in Taiwan as a receptor site to collect the aerosol samples, with the emphasis on the potential long range transport. More than 300 filter samples were collected and analyzed. The authors found that much higher As occurred in Spring compared to other seasons, which was associated with the intensive biomass burning in South Asia. However, the biomass burning in SouthEast Asia (i.e. Indo-China Peninsula) did not release much As. In general, I think this work is well designed. The laboratory analysis (ions and elements), online observation in field (e.g CO) and modelling (WRF-Chem) were integrated from different aspects. And the finding of this work is meaningful for the scientists in the field of atmospheric chemistry and biogeochemical cycling of elements. But there are several

questions still exist in the manuscript and needed to be solved during the revision.

Specific comments: (1) Line 73-74, Actually, besides spring, the intensive biomass burning in India also happened in autumn (from late October to early November), due to the burning of rice (paddy) residue after the harvest. More information could be found in the literature, e.g. Eos, Vol. 95, No. 37, 16 September 2014. Crop Residue Burning: A Threat to South Asian Air Quality. (2) Line 109, Regarding the description of sampling site, it is better to note clearly that it is located in Taiwan. (3) Line 148, what is the recovery of As in the ICP-MS analysis? (4) Line 194, I think there is no need to mention Chongqing here. Maybe a broader geographic area like Sichuan Basin is better. (5) As shown in Fig 8a, during this period, why there is no firespots observed by MODIS in Southeast Asia. And please modify the Hehuanshan into Mount Hehuan in the figures. (6) Line 381, if possible, please provide more details about the usage of lead arsenate(LA) in South Asia, especially in the agricultural sector. What is the total amount of this insecticides used in South Asia every year? (7) In the future, maybe the authors could try to analyze the lead isotope in aerosol samples with high As concentration, to further reveal the source of Pb, as well as its relation to As.

---

## Author Comment (AC1) · 19 Jul 2018

In this work, the authors choose Mount. Hehuan (3000 m asl) in Taiwan as a receptor site to collect the aerosol samples, with the emphasis on the potential long range transport. More than 300 filter samples were collected and analyzed. The authors found that much higher As occurred in Spring compared to other seasons, which was associated with the intensive biomass burning in South Asia. However, the biomass burning in SouthEast Asia (i.e. Indo-China Peninsula) did not release much As. In general, I think this work is well designed. The laboratory analysis (ions and elements), online observation in field (e.g CO) and modelling (WRF-Chem) were integrated from different aspects. And the finding of this work is meaningful for the scientists in the field of atmospheric chemistry and biogeochemical cycling of elements. But there are several

questions still exist in the manuscript and needed to be solved during the revision.

Specific comments: (1) Line 73-74, Actually, besides spring, the intensive biomass burning in India also happened in autumn (from late October to early November), due to the burning of rice (paddy) residue after the harvest. More information could be found in the literature, e.g. Eos, Vol. 95, No. 37, 16 September 2014. Crop Residue Burning: A Threat to South Asian Air Quality. (2) Line 109, Regarding the description of sampling site, it is better to note clearly that it is located in Taiwan. (3) Line 148, what is the recovery of As in the ICP-MS analysis? (4) Line 194, I think there is no need to mention Chongqing here. Maybe a broader geographic area like Sichuan Basin is better. (5) As shown in Fig 8a, during this period, why there is no firespots observed by MODIS in Southeast Asia. And please modify the Hehuanshan into Mount Hehuan in the figures. (6) Line 381, if possible, please provide more details about the usage of lead arsenate(LA) in South Asia, especially in the agricultural sector. What is the total amount of this insecticides used in South Asia every year? (7) In the future, maybe the authors could try to analyze the lead isotope in aerosol samples with high As concentration, to further reveal the source of Pb, as well as its relation to As.

[Figure]

[Figure]

This paper presents one year of daily TSP samples at Mount Hehuan, Taiwan. The samples were analysed for inorganic ions and trace metals and the analysis is supplemented by CO concentration measurements and air mass history analysis.

The manuscript concentrates on the observation of elevated As concentration in biomass burning (BB) plumes. As is found to correlate strongly with Pb, and based on As/Pb ratio of approx. 0.2 the authors suggest that the elevated As concentrations in BB plumes may originate in the usage of lead arsenate pesticide.

While the observation of elevated As in BB plumes is interesting, the concentrations are well below air quality limits at Mount Hehuan. Also the estimated As emission of

0.17 tons per year in South Asian BB is very small (approx. 0.5%) compared to the annual global emissions of 31 tons. Therefore, the results are of only limited interest.

In my opinion this data set requires a more thorough statistical analysis to merit publication in ACP. Please apply principal component analysis (PCA), or a similar factorisation technique, to the data to see if other sources than biomass burning can be identified. Please provide also average concentrations of trace metals and inorganic ions for the different air mass origins you defined. I believe this would provide a valuable reference point for East Asian free troposphere.

Minor comments

L30-35 "Finally, the net influence of BB activities on airborne As concentrations has been simply estimated by comparing the differences of As concentrations between BB and non-BB days. The result showed, on average, the contribution of BB activities over S Asia to airborne As was approximately 1.0 ng m-3, which accounted 63% for total airborne As concentrations in the springtime."

Do you mean that As concentration was on average 1.0 ng m-3 higher during BB days than non-BB days in the springtime? This is quite a different conclusion than "the contribution of BB activities over S Asia to airborne As was approximately 1.0 ng m-3". Please define springtime and state the BB contribution to As on annual level.

L36-38 "Using this value, arsenic emissions from S Asian BB activities were estimated to be 0.17 tons yr-1, causing extremely high airborne As concentrations over the sub-tropical free troposphere, and impacted As cycles on a regional scale."

I wouldn't call the concentration "extremely high" as it is below most (if not all) air quality limits. It is high for free troposphere, but much higher average values have been reported from continental boundary layer. Also, 0.17 tons yr-1 is approx. 0.5% of the global annual emissions, so not a huge amount. However, As in BB smoke may pose a serious health risk close to the fire, where the smoke is not yet diluted.

[Figure]

L60-63 "However, whether BB-derived As can be traveled to long distance and influenced As cycles at its downstream regions is still an open question."

Wai et al. (2016) show that As in general can travel long distances globally. Also it is well known that BB smoke can travel long distances. So it is not a surprise that As released in BB can travel long distances.

L64-65 "volatile organic carbon (VOC) and particulate matters (PM))" Do you mean "volatile organic compounds (VOCs) and particulate matter (PM)" ?

L69 Do you mean "Kondo et al., 2004"? Please check all occurrences of this reference.

L75-77 Most BB smoke is emitted within boundary layer (e.g. Val Martin et al., 2010), though some is of course lifted into free troposphere.

L176 Please explain shortly how the clustering was done. Did you take into account the height of the trajectory?

L209 Please explain which parameters/fields from WRF-Chem did you use in this study. Did you obtain emission sensitivity from the model? Did you take deposition (dry and/or wet) into account?

L231 Are concentrations presented under prevailing conditions or in STP or NTP?

L246-247 "Increased As concentrations coincided with CO peaks on some days, showing some highly anthropogenic As plumes passed over this site." Please explain what you mean with "highly anthropogenic As plumes".

L279-286 Please define the enrichment factor in section 2, it belongs to methods.

L287 and Fig. 5. I don't see much difference between the different seasons in Fig. 5. I think that this is one place where principal component analysis (or similar) would be very useful to differentiate between crustal origins and other sources. See e.g. the analysis by Venter et al. (2017) on trace metal concentrations.

[Figure]

L309-313 "As mentioned above, BB activities may be an important regionally source for high As concentrations over the subtropical free troposphere, especially during the spring period; consequently, in this section, we prove the hypothesis using backward trajectory analyses and MODIS fires observations together with WRF-Chem model simulated results."

This is quite a strong statement. Given the uncertainties in trajectory calculations, I think the correlation with K+ is a stronger indicator of BB origin of As. Please consider re-phrasing.

L313 Please state (with appropriate references) which MODIS product you mean by "fire spots". It would be good to include a sub-section in section 2 detailing which methods are used to characterise BB plumes.

L323-326 "For convenience, prior to further analysis we arbitrarily chose a K+ concentration of 109 ng m-3 (the 25th percentile value of potassium ion) as a criterion value for identifying the suspected BB event. A second criterion (CO concentration up to 160 ppb) was also added for selection of the BB plume."

Do you mean that your site is within a BB plume 75% of time? Using 25th percentile value of K+ as threshold sounds unrealistic to me. At the same time using CO threshold of <160ppb sounds very strange. Please explain.

L332-334 Please plot back-trajectories for high-As BB-plumes and low-As BB-plumes on a map separately (e.g. with different colours). Is there a difference in the footprint area? The back-trajectory clusters have substantial overlap.

Here it would be better to define high and low As plumes based on As/CO or As/K ratio. The absolute As concentration in a plume depends strongly on dilution, whereas the above ratios should remain rather constant independent of dilution. Better still to use excess (delta) As, CO and K+ to calculate the ratios.

It would be interesting to see if principal component analysis finds a factor with high As

and Pb contributions. Inspecting the air mass history of such a factor should make it very clear where the As plumes originate.

L373-378 What is the As/Pb ratio in the ground water in this region? Is it significantly different from the ratio of 0.22 for lead arsenate?

L428-429 Is the difference calculated correct here? Also, this difference depends mostly on BB plume dilution.

L433-436 Please try to find BB-plume As factor with PCA. That will give you a more reliable estimate of the BB-contribution to As load at Mount Hehuan.

L438 How do you define delta? Is this excess concentration in the plume? This should be introduced already in section 2.

L438-442 Please indicate the unit of delta As / delta K+ and delta As / delta CO.

L444-445 For an order of magnitude estimate a 15-year old emission inventory is probably ok, but you may wish to use a newer inventory. At least GFED4 (van der Werf et al., 2017) differentiates between agricultural residue burning and other kind of fires.

L457-458 "Backward trajectory and WRF-Chem model proved that the high As plumes originated mainly from S Asia."

The high As plumes may have originated from S Asia, but in my opinion this has not been proved. Especially Fig. 8, which is the only place where WRF-Chem is utilised, is misleading and poorly documented. Please see comments on Fig. 8 below.

L461-463 "Furthermore, we roughly estimated that approximately 1.0 ng m-3 of As was contributed by biomass burning activities over the South Asian continent, accounting 63% of total airborne As in the springtime."

Here "1.0 ng m-3 of As" sounds strange considering that you present only annual mean (0.5 ng m-3) before. Please clarify and state the BB contribution to As on annual level and define "springtime".

[Figure]

Fig. 6 Please explain how the colour coding (As bins) is done. Now it seems to me that "<5th" refers to highest 5 % of values, though usually 5th percentile would include the lowest 5 % of data points.

Fig.8 Caption: "Figure 8 MODIS fires and WRF-Chem modeled results of BB plumes on (a) April 3 and (b) March 25." Please check if panel (b) presents March 25 or March 15. The "fire spot" legend in the plot suggests March 15 to me.

Please indicate what the blue arrows and the shaded areas represent.

My main concern with this figure is that for panel (a) all fire observations outside India are left out. This gives a very misleading picture of the potential sources of the BB plumes for April 3 observations at Mount Hehuan, as during March 25 – April 3 there are much more fires between India and Taiwan than in India (see attached screenshot from FIRMS).

Also Fig. 8b the "fire spots" are drawn only for a small sub-region of the map.

Finally, there are still quite a few language issues, which should be carefully checked. Please also consider splitting some very long paragraphs into shorter ones (e.g. L166-207 is all just one paragraph).

From the supplement: Please provide proper caption for all figures. Please indicate also here the proper details of MODIS "fire spots".

[Figure]

[Figure]

**Fig. 1.**
Response to reviewer's comments (Manuscript No. ACP-2018-108)

**Response to reviewer's comments**

**(Manuscript No. ACP-2018-108)**

**Reviewer #1**

In this work, the authors choose Mount. Hehuan (3000 m asl) in Taiwan as a receptor site to collect the aerosol samples, with the emphasis on the potential long range transport. More than 300 filter samples were collected and analyzed. The authors found that much higher As occurred in Spring compared to other seasons, which was associated with the intensive biomass burning in South Asia. However, the biomass burning in South East Asia (i.e. Indo-China Peninsula) did not release much As. In general, I think this work is well designed. The laboratory analysis (ions and elements), online observation in field (e.g CO) and modelling (WRF-Chem) were integrated from different aspects. And the finding of this work is meaningful for the scientists in the field of atmospheric chemistry and biogeochemical cycling of elements. But there are several questions still exist in the manuscript and needed to be solved during the revision.

**1st comment**

Line 73-74, Actually, besides spring, the intensive biomass burning in India also happened in autumn (from late October to early November), due to the burning of rice (paddy) residue after the harvest. More information could be found in the literature, e.g. Eos, Vol. 95, No. 37, 16 September 2014. Crop Residue Burning: A Threat to South Asian Air Quality

**Author's response:**

Thanks for the reviewer's comment. We agree that some fire spots were observed in autumn in India. The biomass burning activities in autumn were probably caused by burning of rice residues. As reported by Pochanart et al. (2003), the fire spots were much less in autumn season compared to those in spring. On the other hands, the seasonal variations of fire spots observed by MODIS during the sampling period are plotted in Figure S2. As seen, the fire spots over S Asia in the autumn were nearly 2100 which was only 20 % of those (~10000) in the springtime. As a result, significant biomass burning activities over S Asia also maximizes in spring season.

**2nd comment**

Line 109, Regarding the description of sampling site, it is better to note clearly that it is located in Taiwan.

**Author's response:**

In the revised manuscript, we have clearly stated that Mount Hehuan is located in Taiwan. (lines 115 and 116 on page 5)

**3rd comment**

Line148, what is the recovery of As in the ICP-MS analysis?

**Author's response:**

The recovery and precision of As were 106 % and 2 %, respectively. (lines 155 and 156 on page 7)

**4th comment**

Line 194, I think there is no need to mention Chongqing here. Maybe a broader geographic area like Sichuan Basin is better.

**Author's response:**

As suggested, we have replaced "Chongqing" by "Sichuan Basin". (line 222 on page 9)

**5th comment**

As shown in Fig 8a, during this period, why there is no fires pots observed by MODIS in Southeast Asia. And please modify the Hehuanshan into Mount Hehuan in the figures.

**Author's response:**

Thanks for the reviewer's comment. We have added all the MODIS fires in study domain in Figs. 7a and 7b in the revised manuscript. On the other hand, we have deleted "Hehuanshan" in this figure. (on page 39)

**6th comment**

Line 381, if possible, please provide more details about the usage of lead arsenate (LA) in South Asia, especially in the agricultural sector. What is the total amount of this insecticides used in South Asia every year?

**Author's response:**

Thanks for the reviewer's comment. In the revised manuscript, we have added the information of LA as "Lead arsenate (LA, $[Pb_5OH(AsO_4)_3]$; As/Pb~0.22) was the most extensively used as the arsenical insecticides in the world. It was used as an insecticide for gypsy moths invading hardwood forests in 1892. LA can be adhered to the surfaces of plants. Although LA was officially banned as insecticide in 1990's in many developed countries, but has not been banned in India nowadays" in lines 444 – 449 on page 18. Unfortunately, we can't obtain the total amount of this insecticide they used.

**7th comment**

In the future, maybe the authors could try to analyze the lead isotope in aerosol samples with high As concentration, to further reveal the source of Pb, as well as its relation to As.

**Author's response:**

Thanks for the reviewer's comment. Lead isotope ratios are good tools to track the potential sources and identify the long-range transported particulate lead in the troposphere. This is a good research direction which we can develop in the future.

**Response to reviewer's comments**

**(Manuscript No. ACP-2018-108)**

**Reviewer #2**

This paper presents one year of daily TSP samples at Mount Hehuan, Taiwan. The samples were analysed for inorganic ions and trace metals and the analysis is supplemented by CO concentration measurements and air mass history analysis. The manuscript concentrates on the observation of elevated As concentration in biomass burning (BB) plumes. As is found to correlate strongly with Pb, and based on As/Pb ratio of approx. 0.2 the authors suggest that the elevated As concentrations in BB plumes may originate in the usage of lead arsenate pesticide. While the observation of elevated As in BB plumes is interesting, the concentrations are well below air quality limits at Mount Hehuan. Also the estimated As emission of 0.17 tons per year in South Asian BB is very small (approx. 0.5%) compared to the annual global emissions of 31 tons. Therefore, the results are of only limited interest. In my opinion this data set requires a more thorough statistical analysis to merit publication in ACP. Please apply principal component analysis (PCA), or a similar factorization technique, to the data to see if other sources than biomass burning can be identified. Please provide also average concentrations of trace metals and inorganic ions for the different air mass origins you defined. I believe this would provide a valuable reference point for East Asian free troposphere.

**Author's response:**

Thanks for the reviewer's comment. In the revised manuscript, we have added the results of principle component analysis (PCA) to qualitatively identify the potential sources of airborne TSP observed at Mount Hehuan (see the author's response to the 12[th] comment and 19[th] comment). We have also added the concentrations of trace metals and inorganic ions in the different air clusters (see in Table 1 and lines 333 - 350

on page 14 and lines 351-352 on page 15).

**1st comment**

L30-35 "Finally, the net influence of BB activities on airborne As concentrations has been simply estimated by comparing the differences of As concentrations between BB and non-BB days. The result showed, on average, the contribution of BB activities over S Asia to airborne As was approximately 1.0 ng m-3, which accounted 63% for total airborne As concentrations in the springtime." Do you mean that As concentration was on average 1.0 ng m-3 higher during BB days than non-BB days in the springtime? This is quite a different conclusion than "the contribution of BB activities over S Asia to airborne As was approximately 1.0 ng m-3". Please define springtime and state the BB contribution to As on annual level.

**Author's response:**

Thanks for the reviewer's comment. In the revised manuscript, we have re-organized the sentence and deleted the word of "contribution". The net influence of S Asian BB activities on airborne As concentrations has been estimated by comparing the differences of As concentrations between BB and non-BB days. On average, the difference of As concentrations on the BB and non-BB days was 1.0 ng $m^{-3}$, which accounted 63 % for the average As concentration on BB days during the S and SE Asian BB periods (lines 39-41 on page 2). This finding indicated that S Asian BB activities was a dominant source for high As concentrations during the S and SE Asian BB seasons.

**2nd comment**

L36-38 "Using this value, arsenic emissions from S Asian BB activities were estimated to be 0.17 tons yr-1, causing extremely high airborne As concentrations over the subtropical free troposphere, and impacted As cycles on a regional scale." I wouldn't call the concentration "extremely high" as it is below most (if not all) air quality limits. It is high for free troposphere, but much higher average values have been reported from continental boundary layer. Also, 0.17 tons yr-1 is approx. 0.5% of the global annual emissions, so not a huge amount. However, As in BB smoke may pose a serious health risk close to the fire, where the smoke is not yet diluted.

**Authors response:**

Thanks for the reviewer's comment. In the revised manuscript, we have re-organized the abstract and deleted "extremely high" in the abstract.

**3rd comment**

L60-63 "However, whether BB-derived As can be traveled to long distance and influenced As cycles at its downstream regions is still an open question." Wai et al. (2016) show that As in general can travel long distances globally. Also it is well known that BB smoke can travel long distances. So it is not a surprise that As released in BB can travel long distances.

**Authors response:**

In the revised manuscript, we have rephrased the sentence of "However, whether As could be……." to "However, the influence of …….is well not understood". (lines 67 and 68 on page 3)

**4th comment**

L64-65 "volatile organic carbon (VOC) and particulate matters (PM))" Do you mean "volatile organic compounds (VOCs) and particulate matter (PM)" ?

**Author's response:**

That is a typo. We have corrected "volatile organic carbon (VOC)" to "volatile organic compounds (VOCs)" in lines 70 and 71 on page 3.

**5th comment**

L69 Do you mean "Kondo et al., 2004"? Please check all occurrences of this reference.

**Author's response:**

Thanks for the reviewer's comment. We have corrected "Kondo et al., 2003" to "Kondo et al., 2004" (line75 on page 3 and line 85 on page 4) and checked all occurrences of this reference.

**6th comment**

L75-77 Most BB smoke is emitted within boundary layer (e.g. Val Martin et al., 2010), though some is of course lifted into free troposphere.

**Author's response:**

Thanks for the reviewer's comment. We agree that most BB smoke is emitted within boundary layer, though some is of course lifted into free troposphere (lines 81-85 on page 4). In the revised manuscript, we have added the reference done by Val Martin et al. (2010) in line 85 on page 4 and in references lists on page 27.

**7th comment**

L176 Please explain shortly how the clustering was done. Did you take into account the height of the trajectory?

**Author's response:**

Thanks for the reviewer's comment. To identify the potential sources of particulate arsenic observed at Mount Hehuan, five-day backward trajectory starting at 3000 m a.s.l. were computed at 12:00 LT once every day with a time step of 6 hours. Most air parcels started from the originated regions with the altitude of approximately 6000 m and then descended to the receptor site. During the sampling period, a total of 1865 backward trajectories were computed during the sampling periods. According the originated regions of air parcels, we divided the trajectories into five groups, namely, Northern China (NC), Pacific Ocean (PO), South Sea (SS), Southeast Asia (SEA) and South Asia (SA). (lines 203-206 on page 9)

**8th comment**

L209 Please explain which parameters/fields from WRF-Chem did you use in this study. Did you obtain emission sensitivity from the model? Did you take deposition (dry and/or wet) into account?

**Author's response:**

As suggested, we have added the brief introduction of parameters from WRF-Chem model in the revised manuscript. In this study, a tracer module in WRF-Chem developed by Lin et al (2009) was employed to identify the transport of BB plumes. This model has been successfully simulated and identified the biomass burning transportation from S and SE Asia (Chi et al. 2010; Lin et al. 2009; 2014). The tracers were assigned to the fire locations derived from MODIS satellite data over the study domain. They were placed at the first level above the surface at each fire location with a concentration of 1 unit per day. The dry and wet deposition functions are considered in the model. (lines 238-246 on page 10)

**9th comment**

Are concentrations presented under prevailing conditions or in STP or NTP?

**Author's response:**

Thanks for the reviewer's comment. All the concentrations are presented under prevailing conditions. (lines 273 and 274 on page 11)

**10[th] comment**

L246-247 "Increased As concentrations coincided with CO peaks on some days, showing some highly anthropogenic As plumes passed over this site." Please explain what you mean with "highly anthropogenic As plumes"

**Author's response:**

Thanks for the reviewer's comment. CO is a good indicator of anthropogenic emissions. As investigated by streets et al. (2003), the annual emissions of CO in Asia was 279 Tg. Apart from vehicle emissions, coal use, fuel combustion and industries were also important sources for CO. In this work, we found some increased As coincide with enhanced CO, but $K^+$ (a tracer for biomass burning) concentrations did not increase. This indicates that As was not contributed by biomass burning. However, we can't identify the potential source for As in these high CO events. Consequently, in the revised manuscript, we have omitted the sentence.

**11[th] comment**

L279-286 Please define the enrichment factor in section 2, it belongs to methods.

**Author's response:**

In the 12th comment, the reviewer thought PCA is better to identify the aerosol sources than enrichment factor (EF) analysis. Thus, we have deleted all parts relevant to EF analysis in the revised manuscript.

**12[th] comment**

L287 and Fig. 5. I don't see much difference between the different seasons in Fig. 5. I think that this is one place where principal component analysis (or similar) would be very useful to differentiate between crustal origins and other sources. See e.g. the analysis by Venter et al. (2017) on trace metal concentrations.

**Author's response:**

As suggested, the enrichment factor analysis has been replaced by principal component analysis in the revised manuscript. (lines 172-175 on page 7, lines 176-190 on page 8, lines 333-350 on page 14 and lines 351-352 on page 15 along with Table 1)

**13th comment**

L309-313 "As mentioned above, BB activities may be an important regionally source for high As concentrations over the subtropical free troposphere, especially during the spring period; consequently, in this section, we prove the hypothesis using backward trajectory analyses and MODIS fires observations together with WRF-Chem model simulated results". This is quite a strong statement. Given the uncertainties in trajectory calculations, I think the correlation with K+ is a stronger indicator of BB origin of As. Please consider re-phrasing.

**Author's response:**

Thanks for the reviewer's comment. Actually, we did correlation analysis between As and $K^+$ for different As bin values. As shown in Figure 5, significant correlations ($r = 0.78$, $p < .05$ for the 95th percentile value of As) between As and $K^+$ when high As concentrations occurred. Since $K^+$ is a good indicator for biomass burning; and therefore, high As concentrations might be emitted from BB activities. (353-356 on page 15)

**14th comment**

L313 Please state (with appropriate references) which MODIS product you mean by "fire spots". It would be good to include a sub-section in section 2 detailing which methods are used to characterise BB plumes.

**Author's response:**

As suggested, we have added the section 2.6 to give the introduction of MODIS fire spots we obtained during the sampling period. (lines 254-264 on page 11)

**15th comment**

L323-326 "For convenience, prior to further analysis we arbitrarily chose a K+ concentration of 109 ng m-3 (the 25th percentile value of potassium ion) as a criterion value for identifying the suspected BB event. A second criterion (CO concentration up to 160 ppb) was also added for selection of the BB plume." Do you mean that your site is within a BB plume 75% of time? Using 25th percentile value of K+ as threshold sounds unrealistic to me. At the same time using CO threshold of <160ppb sounds very strange. Please explain.

**Author's response:**

Thanks for the reviewer's comment. Actually, we chose a $K^+$ concentrations of > 109 ng $m^{-3}$ (75th percentile value) and a CO concentration of > 160 ppb (75th percentile value) as threshold values to identify the suspected BB samples. The 75th percentile values for both species can be representative of high $K^+$ and CO conditions and used to select the suspected BB samples. In the revised manuscript, we have corrected these sentences (lines 376-379 on page 15). Meanwhile, we have also corrected the statements in Figure 4a (on page 36).

**16th comment**

L332-334 Please plot back-trajectories for high-As BB-plumes and low-As BB-plumes on a map separately (e.g. with different colours). Is there a difference in the footprint area? The back-trajectory clusters have substantial overlap. Here it would be better to define high and low As plumes based on As/CO or As/K ratio. The absolute As concentration in a plume depends strongly on dilution, whereas the above ratios should remain rather constant independent of dilution. Better still to use excess (delta) As, CO and K+ to calculate the ratios. It would be interesting to see if principal component analysis finds a factor with high As and Pb contributions. Inspecting the air mass history of such a factor should make it very clear where the As plumes originate.

**Author's response:**

Thanks for the reviewer's comment. As suggested, we have added the plot to illustrate backward trajectories for the cases of high-As and low-As plumes on a map separated with different colors as seen in Figure S3. On the other hand, we agreed the reviewer's comment that the absolute As concentration might be diluted during their transport. We also calculate the As/CO and As/K$^+$ for the all data sets. Unfortunately, As did not correlate well with As/K$^+$ (R$^2$ = 0.03, p >.05). This might be explained by the different/additional emission sources for the two species picked up by air masses during their long-range transport, especially during the periods between July to December. This also reflect that the ratios of As/K$^+$ may not suitable to identify the high-/low As plumes. Thus, we still use the absolute As concentrations to identify the high As events and attempt to investigate their potential sources. Moreover, we also use PCA to check the As sources over Mount Hehuan. The result showed that a high loading of K$^+$ (0.71) and a moderate loading of CO (0.50) in the PC2 during the S and SE Asian BB periods, indicating BB origins. Meanwhile, a moderate of As (0.67) was also found in this factor. This implies that As was mainly from BB activities during the S and SE Asian BB seasons. (lines 333-350 on page 14 and lines 351-352 on page 15)

**17th comment**

L373-378 What is the As/Pb ratio in the ground water in this region? Is it significantly different from the ratio of 0.22 for lead arsenate?

**Author's response:**

Thanks for the reviewer's comment. We have collected some papers about distributions of trace metals in ground water over this region (Ali et al., 2016, Environmental Nanotechnology, Monitoring and Management; Islam et al., 2017, Marine Pollution Bulletin; Islam et al., 2017, Chemosphere). The As/Pb ratios in ground water over this region are not constant levels. They range widely from 0.04 to 0.22 in the ground water and from 0.13 to 0.58 in the sediments of rivers over this region. However, the As/Pb ratio of LA is within the range of these values.

**18th comment**

L428-429 Is the difference calculated correct here? Also, this difference depends mostly on BB plume dilution.

**Author's response:**

Thanks for the reviewer's comment. We agree the reviewer's comment, that is, the difference depends mostly on the dilution of BB plume. However, based on the observed data, we can only use the difference of As concentrations between BB and non-BB days to roughly estimate the net influence of BB activities over S and SE Asia on arsenic concentrations in the subtropical free troposphere through uncertainties were existed in the estimations (Kato et al., 2002; Lin et al., 2009).

**19th comment**

L433-436 Please try to find BB-plume As factor with PCA. That will give you a more reliable estimate of the BB-contribution to As load at Mount Hehuan.

**Author's response:**

As suggested, we have added PCA analysis in sections 2.3 and 3.2 in the revised manuscript. For the PCA results, we can clearly see that BB activity was one of the major sources of airborne TSP at Mount Hehuan site during the S and SE Asian biomass burning seasons (from January to May). The explained variance of this factor was approximately 26 %. Interestingly, moderate loadings of As and Pb were also found in this factor, indicating that As and Pb were from BB activities. In contrast, high loadings of As and Se were found in PC 2 (explained variance was 17 %) during the non-BB periods (from June to December), suggesting that As was mainly from coal-combustion. (lines 333-350 on page 14 and lines 351-352 on page 15 along with Table 1)

**20th comment**

L438 How do you define delta? Is this excess concentration in the plume? This should be introduced already in section 2.

**Author's response:**

We roughly identified $\Delta K^+$, $\Delta CO$ and $\Delta As$ as the differences of concentrations in $K^+$, As and CO between BB and non-BB days in S and SEA air clusters. Subsequently, we can obtain the ratios of $\Delta K^+/\Delta CO$ and $\Delta As/\Delta CO$ and estimate the $K^+$ or As emissions from BB activities over S and SE Asia. In our estimation, the emission rates of As from BB activities over S Asia was 0.17 tons per year. (lines 499-500 on page 20 and lines 501-505 on page 21)

**21th comment**

L438-442 Please indicate the unit of delta As / delta K+ and delta As / delta CO.

**Author's response:**

Thanks for the reviewer's comment. To obtain the $\Delta K^+/\Delta CO$ and $\Delta As/\Delta CO$, we first convert the units of $K^+$ and As from $ng\ m^{-3}$ to ppb based on the airborne temperature at Mount Hehuan and molecular weights of $K^+$ (39) and As (75). Thus, we can obtain the $\Delta K^+/\Delta CO$ and $\Delta As/\Delta CO$ without units. (lines 499-500 on page 20 and line 501-503 on page 21)

**22th comment**

L444-445 For an order of magnitude estimate a 15-year old emission inventory is probably ok, but you may wish to use a newer inventory. At least GFED4 (van der Werf et al., 2017) differentiates between agricultural residue burning and other kind of fires.

**Author's response:**

As suggested, we have cited the paper published by van der Werf et al. (2017) in the revised manuscript (lines 78-79 on page 3) and reference lists. van der Werf et al. (2017) quantified the global fire emission patterns during 1997 – 2016. They also estimated the carbon and CO emissions from burned activities for different regions during 1997-2011 (GFED3) and 1997-2016 (GFED4), and compared their differences. Unfortunately, they did not separate the CO emissions produced by biomass burning over Indian Subcontinent from Southeast Asia (including Indian Subcontinent and Indo-China Peninsula). Thus, the arsenic emissions (calculated by $\triangle K^+/\triangle CO$) from biomass burning over S Asia were still calculated based on the CO emission data investigated by Stresst et al. (2003) in this work.

**23th comment**

L457-458 "Backward trajectory and WRF-Chem model proved that the high As plumes originated mainly from S Asia." The high As plumes may have originated from S Asia, but in my opinion this has not been proved. Especially Fig. 8, which is the only place where WRF-Chem is utilised, is misleading and poorly documented. Please see comments on Fig. 8 below.

**Author's response:**

Thanks for the reviewer's comment. In the revised manuscript, we have re-plotted this figure (new Figure 7) and the details are seen in the response to 26th comment.

**24th comment**

L461-463 "Furthermore, we roughly estimated that approximately 1.0 ng m-3 of As was contributed by biomass burning activities over the South Asian continent, accounting 63% of total airborne As in the springtime." Here "1.0 ng m-3 of As" sounds strange considering that you present only annual mean (0.5 ng m-3) before. Please clarify and state the BB contribution to As on annual level and define "springtime".

**Author's response:**

Thanks for the reviewer's comment. The same response can be seen in 1st comment.

**25th comment**

Fig. 6 Please explain how the colour coding (As bins) is done. Now it seems to me that "<5th" refers to highest 5 % of values, though usually 5th percentile would include the lowest 5 % of data points.

**Author's response:**

Thanks for the reviewer's comment. We agree that 5th percentile value would include the lowest 5% of data points. In the revised manuscript, we have corrected the statements of legend in Figure 4 and also corrected all mistakes throughout the paper. (lines 301-302 on page 13, lines 327 on page 14 and lines 355 and 359 on page 15, and line 377 on page 16)

**26ᵗʰ comment**

Fig.8 Caption: "Figure 8 MODIS fires and WRF-Chem modeled results of BB plumes on (a) April 3 and (b) March 25." Please check if panel (b) presents March 25 or March15. The "fire spot" legend in the plot suggests March 15 to me. Please indicate what the blue arrows and the shaded areas represent. My main concern with this figure is that for panel (a) all fire observations outside India are left out. This gives a very misleading picture of the potential sources of the BB plumes for April 3 observations at Mount Hehuan, as during March 25 – April 3 there are much more fires between India and Taiwan than in India (see attached screenshot from FIRMS). Also Fig. 8b the "fire spots" are drawn only for a small sub-region of the map.

**Author's response:**

Thanks for the reviewer's comment. We have re-plotted the new Figure 7 in the revised manuscript. In this figure, we have added all the fire spots observed by MODIS in the study domain ranging from 5 to 40 ˚N and 65 to 135 ˚E. In this figure, we show the simulated results by WRF-Chem model on (a) April 3 and (b) March 15. In Figure 7 (a), extensive fire spots were observed over India-subcontinent, Indo-China Peninsula and Southern China from March 25 to April 2. As computed by HYSPLIT model, the air parcels were mainly from Indian Subcontinent (see in Figure S4a), and therefore the tracers were assigned to the fire locations derived from MODIS satellite data over Indian Subcontinent ranging from 5 to 38 ˚N and 65 to 90 ˚E and they were placed at the surface level above the surface at each fire location with the concentration of a unit per day. The result showed that the significant BB plume originated over burned areas, transporting to east direction, and passed over Mount Hehuan (lines 395-400 on page 16 and lines 401-402 on page 17). In Figure 7 (b), the tracers were placed at the surface level above the surface at each fire location in the Indo-China Peninsula ranging from 5 to 30 ˚N and 90 to 110 ˚E since the backward trajectories originated mainly from

Indo-China Peninsula. As seen, the WRF-Chem model showed that the significant tracer concentration laid in northeast-southwest belt and covered Taiwan on March 15.

(lines 411-416 on page 17)

**27th comment**

Finally, there are still quite a few language issues, which should be carefully checked.

Please also consider splitting some very long paragraphs into shorter ones (e.g. L166- is all just one paragraph).

**Author's response:**

Thanks for the reviewer's comment. In the revised manuscript, we have checked the language and split the very long paragraph shorter ones.

**28th comment**

From the supplement: Please provide proper caption for all figures. Please indicate also here the proper details of MODIS "fire spots".

**Author's response:**

Thanks for the reviewer's comment. We have provided proper captions for all figures and details of MODIS "fire spots" in supporting materials.

**Changes are marked by red color**

[revised manuscript text omitted]

**(a)**

[Figure]

**(b)**

[Figure]

**Figure 7**

[Figure]

**Figure 8**

[Figure]

**Figure 9**

**Supplementary Materials**

**Enhancements of Airborne Particulate Arsenic over the Subtropical Free Troposphere: Impact by South Asian Biomass Burning**

**Yu-Chi Lin[1,2,3], Shih-Chieh Hsu[3], Chuan-Yao Lin[3], Shuen-Hsin Lin[3], Yi-Tang Huang[3], Yunhua Chang[1,2], Yan-Lin Zhang[1,2*]**

[1.] *Yale-NUIST Center on Atmospheric Environment, Nanjing University of Information Science and Technology, Nanjing, Jiangsu, China.*

[2.] *Key Laboratory of Meteorological Disaster, Ministry of Education & Collaborative Innovation Center on Forecast and Evaluation of Meteorological Disasters, Nanjing University of Information Science and Technology, Nanjing, Jiangsu, China.*

[3.] *Research Center for Environmental Changes (RCEC), Academia Sinica, Taipei, Taiwan, R.O.C.*

*Corresponded to Yan-Lin Zhang (*zhangyanlin@nuist.edu.cn; dryanlinzhang@outlook.com*)*

This "Supplementary Materials" contains one table and five figures. Table S1 lists the average concentrations of chemical species in TSP samples observed at Mount Hehuan in different air clusters. Figure S1 plots time series of daily concentrations of airborne As, Pb and K$^+$ in TSP along with CO observed at Mount Hehuan from September 2011 to September 2012. Figure S2 reveals monthly distributions of MODIS fire spots observed over southeast (SE) Asia and south (S) Asia from September 2011 to September 2012. Figure S3 shows five-day backward trajectories observed at Mount Hehuan in different BB cases. Figure S4 five-day backward trajectory at Mount Hehuan from (a) March 25 to April 3, 2012 and (b) March 8 to 15, 2012. Figure S5 illustrates the scattered plots of As against Al observed at Mount Hehuan in (a)SA, (b)SEA and (c)other air groups during the SE and S Asian biomass burning seasons.

Table S1 The average concentrations of chemical species in TSP samples observed at

Mount Hehuan in different air clusters. The units of all species are in ng m$^{-3}$.

| | NC | PO | SS | SEA | SA |
|---|---|---|---|---|---|
| Al | 178.9 | 145.0 | 53.6 | 145.9 | 295.8 |
| Fe | 117.8 | 82.2 | 37.0 | 94.6 | 203.5 |
| Na | 89.9 | 62.1 | 49.1 | 88.1 | 116.1 |
| Mg | 46.4 | 20.1 | 13.7 | 35.8 | 94.4 |
| K | 136.8 | 87.0 | 49.3 | 147.4 | 223.6 |
| Ca | 139.1 | 77.6 | 43.8 | 86.3 | 252.9 |
| Sr | 1.0 | 0.5 | 0.3 | 0.7 | 2.3 |
| Ba | 2.5 | 2.0 | 1.0 | 1.7 | 3.0 |
| Ti | 11.4 | 7.2 | 3.6 | 9.3 | 20.6 |
| Mn | 4.7 | 2.1 | 1.1 | 3.0 | 6.4 |
| Co | 0.1 | 0.1 | 0.0 | 0.0 | 0.1 |
| Ni | 0.6 | 0.6 | 0.2 | 0.4 | 0.6 |
| Cu | 27.0 | 15.6 | 20.0 | 21.8 | 20.9 |
| Zn | 18.2 | 12.3 | 6.1 | 8.2 | 13.0 |
| Mo | 0.1 | 0.1 | 0.0 | 0.1 | 0.1 |
| Cd | 0.2 | 0.0 | 0.0 | 0.1 | 0.3 |
| Sn | 0.6 | 0.2 | 0.1 | 0.3 | 0.5 |
| Sb | 0.3 | 0.1 | 0.1 | 0.2 | 0.3 |
| Tl | 0.0 | 0.0 | 0.0 | 0.0 | 0.0 |
| Pb | 5.0 | 1.1 | 0.8 | 2.8 | 7.7 |
| V | 0.6 | 0.3 | 0.2 | 0.6 | 0.8 |
| Cr | 0.9 | 0.9 | 0.5 | 0.5 | 1.0 |
| As | 0.6 | 0.1 | 0.1 | 0.4 | 1.2 |
| Se | 0.2 | 0.0 | 0.0 | 0.1 | 0.1 |
| Ge | 0.0 | 0.0 | 0.0 | 0.0 | 0.0 |
| Rb | 0.6 | 0.3 | 0.2 | 0.5 | 1.0 |
| Cs | 0.0 | 0.0 | 0.0 | 0.0 | 0.1 |
| Ga | 0.2 | 0.1 | 0.0 | 0.1 | 0.2 |
| La | 0.1 | 0.1 | 0.0 | 0.1 | 0.2 |
| Ce | 0.2 | 0.2 | 0.1 | 0.2 | 0.4 |
| Nd | 0.1 | 0.1 | 0.0 | 0.1 | 0.2 |
| P | 51.4 | 32.8 | 28.8 | 32.6 | 29.9 |
| NH$_4^+$ | 659.3 | 53.5 | 143.9 | 663.0 | 1064.5 |
| K$^+$ | 71.2 | 4.5 | 24.5 | 116.6 | 198.8 |
| F$^-$ | 14.8 | 2.6 | 6.7 | 27.6 | 36.4 |
| Cl$^-$ | 84.4 | 16.9 | 79.7 | 126.2 | 163.0 |
| SO$_4^{2-}$ | 1935.5 | 141.2 | 328.4 | 1486.7 | 3049.0 |
| NO$_3^-$ | 490.4 | 48.9 | 188.1 | 882.7 | 1212.9 |

[Figure]

Figure S1 Time series of daily concentrations of airborne As, Pb and K$^+$ in TSP along with CO observed at Mount Hehuan from September 2011 to September

2012.

[Figure]

Figure S2 Monthly distributions of MODIS fire spots observed over southeast (SE) Asia and south (S) Asia. The total fire spots in the figure means the summation of fire spots observed over SE Asia and S Asia. The SE Asia region is identified the Indo-China Peninsula ranging from 5 to 30 $^\circ$N and 90 to 110 $^\circ$E; The S Asia region identified the Indian Subcontinent ranging from 5 to 38 $^\circ$N and 65 to 90 $^\circ$E.

[Figure]

Figure S3 Five-day backward trajectories observed at Mount Hehuan in different BB cases. The trajectories were computed at 12:00 LT (local time) once every day with a time step of 6 hours. The red lines denote the air parcels on Feb. 19, Mar. 30, Mar. 31, Apr. 3, May 5 and 7. 2012 (with high As plumes). The yellow ones represent the air masses from Feb. 25 to 28 and March 15, 2012 (with low As concentrations).

**(a)**

[Figure]

**(b)**

[Figure]

Figure S4 Five-day backward trajectory at Mount Hehuan from (a) March 25 to April 3, 2012 and (b) March 8 to 14, 2012. The trajectories were computed at 12:00 LT (local time) once every day with a time step of 6 hours.

[Figure]

Figure S5 The scattered plots of As against Al observed at Mount Hehuan in (a)SA, (b)SEA and (c)other air groups during the SE and S Asian biomass burning seasons (January to May, 2012).

---

## Author Response (AR2)

**Response to reviewer's comments (ACP-2018-108)**

**Reviwer#1**

Submitted on 13 Aug 2018

Anonymous Referee #2

**Anonymous during peer-review: Yes** No

**Anonymous in acknowledgements of published article: Yes** No

**Recommendation to the editor**

| | |
|---|---|
| **1) Scientific significance**
Does the manuscript represent a substantial contribution to scientific progress within the scope of this journal (substantial new concepts, ideas, methods, or data)? | Excellent **Good** Fair Poor |
| **2) Scientific quality**
Are the scientific approach and applied methods valid? Are the results discussed in an appropriate and balanced way (consideration of related work, including appropriate references)? | Excellent **Good** Fair Poor |
| **3) Presentation quality**
Are the scientific results and conclusions presented in a clear, concise, and well structured way (number and quality of figures/tables, appropriate use of English language)? | Excellent **Good** Fair Poor |

For final publication, the manuscript should be

**accepted as is**

**accepted subject to technical corrections**

accepted subject to **minor revisions**

reconsidered after **major revisions**

I would be willing to review the revised paper, if the editor considers it necessary.

I am **not** willing to review the revised paper.

**rejected**

**Suggestions for revision or reasons for rejection** **(will be published if the paper is accepted for final publication)**

I would like to thank the authors for the additional analysis done. In my opinion the manuscript has improved significantly and can be accepted for publication. There are still some minor language issues (e.g. line 378 "CO concentration up to 160 ppb" should be "CO concentration higher than 160 ppb"), but I trust copy editing will take care of that.

**Authors response:**

We have corrected some minor language mistakes and checked the language through the whole paper.